# ENOUGH IS AS GOOD AS A FEAST: A COMPREHENSIVE ANALYSIS OF HOW REINFORCEMENT LEARNING MITIGATES TASK CONFLICTS IN LLMS

**Zixuan Ren**[1,2] , **Jinliang Lu**[3] , **Junhong Wu**[1,2], **Yang Zhao**[1,2], **Dai Dai**[3] , **Hua Wu**[3],
**Haifeng Wang**[3], **Chengqing Zong**[1,2] *

[1]State Key Laboratory of Multimodal Artificial Intelligence Systems, Institute of Automation, CAS, Beijing, China
[2]School of Artificial Intelligence, University of Chinese Academy of Sciences, Beijing, China
[3]Baidu Inc., Beijing, China
`{renzixuan2021,wujunhong2021,zhaoyang2015}@ia.ac.cn`
`{cqzong}@nlpr.ia.ac.cn`
`{lujinliang,daidai,wu_hua,wanghaifeng}@baidu.com`

## ABSTRACT

Model merging plays a crucial role in consolidating multiple specialized models into a single, unified model, especially in the era of large language models (LLMs). Recent research has primarily focused on developing strategies to enhance merging performance with the trained models, while the impact of training paradigms, such as supervised fine-tuning (SFT) and reinforcement learning (RL), on the effectiveness of model merging remains underexplored. In this study, we systematically explore the merging behavior of RL-trained LLMs compared to those trained with traditional SFT. Through comprehensive evaluations across five representative tasks, we find that RL significantly reduces task conflicts and results in less performance degradation after merging, making RL-trained models particularly well-suited for this process. To unearth the reasons behind the superior suitability of RL for model merging, we conduct extensive empirical experiments and theoretical analyses. Our findings highlight three key factors: (1) On-policy training data in RL control the gradient updates in a smaller magnitude, reducing the risk of overwriting existing knowledge for other tasks in the model. (2) The RL optimization objective, which favors "*enough is as good as a feast*", progressively reduces the magnitude and the number of conflict parameter updates as the model converges. (3) Joint optimization of positive and negative examples in RL steers the model towards an unbiased task-specific parameter subspace, ensuring robust performance while further preventing parameter conflicts.

## 1 INTRODUCTION

Large language models (LLMs) have fundamentally reshaped the landscape of artificial intelligence, capturing growing interest from both academia and industry (Team, 2024; Grattafiori et al., 2024; Xiang et al., 2025). Recent statistics show that there are now more than 270,000 models with over 3 billion parameters available on HuggingFace[1]. These large models often exhibit diverse capabilities and strengths (Shao et al., 2024; Ahmad et al., 2025; Toshniwal et al., 2025), prompting researchers to investigate ensemble techniques that integrate the specialized abilities of different models (Li et al., 2023; Chen et al., 2025; Ruan et al., 2025). Among these techniques, model merging—which directly fuses the parameters of independently fine-tuned models without requiring access to the original training data, expensive retraining, or the maintenance of multiple checkpoints—has emerged as a particularly promising solution (Lu et al., 2024; Yang et al., 2024b).

---

* Corresponding author.
[1]`https://huggingface.co/models`

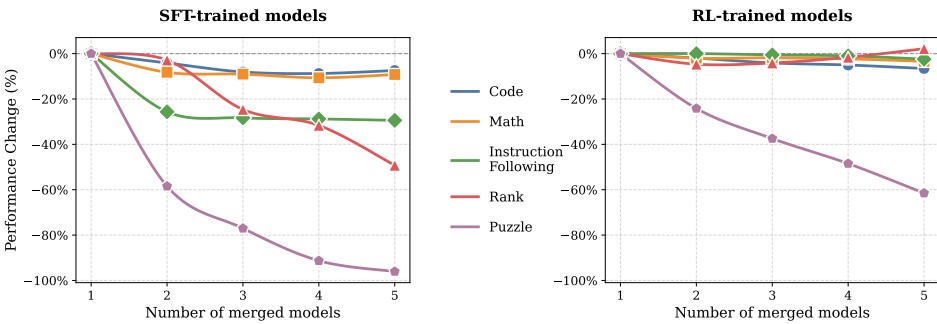

Figure 1: Comparison of performance changes between SFT and RFT in model merging.

A central challenge in model merging is preserving performance when integrating different models, as interference between parameters can lead to performance degradation in the merged model, a phenomenon commonly referred to as *task conflicts*. To mitigate this issue, prior research has proposed a range of model-merging strategies (Ilharco et al., 2023; Yadav et al., 2023; Yu et al., 2024), typically under the assumption that the models involved have already been trained on different tasks. Nonetheless, the approach used to train these task-specific models—which critically impacts the ultimate effectiveness of the merged model—has been largely overlooked. In the era of LLMs, training paradigms can be broadly classified into supervised fine-tuning (SFT) and reinforcement learning (RL) (Hu et al., 2025; DeepSeek-AI et al., 2025). Most existing works focus on merging models trained via SFT, leaving differences between SFT and RL in the context of model merging largely unexplored and deserving of further investigation.

In this work, we investigate the role of fine-tuning paradigms (SFT and RL) in model merging. A series of controlled experiments are conducted across five representative tasks—mathematical reasoning, code generation, instruction following, logical puzzles, and ranking. And we assess the extent to which models fine-tuned with SFT or RL, respectively, preserve their original performance after model merging. Figure 1 reveals that **models trained with RL exhibit significantly better performance preservation after merging than those trained with SFT**. Extensive experiments suggest that our finding holds regardless of different merging methods, various base models, or distinct RL algorithms. Furthermore, we demonstrate that RL alleviates parameter conflicts, thereby making it particularly well-suited for model merging.

To further investigate the reason behind the superiority of RL on model merging, we conduct a series of theoretical and empirical analyses. Our results demonstrate three primary contributing factors: (1) **On-policy training data**: unlike SFT, where training data are sampled from fixed datasets or human annotations, RL relies on data sampled from the model itself. This substantially reduces gradient magnitudes and thereby lowers the risk of overwriting knowledge acquired from other tasks. (2) **The intrinsic dynamics of RL**: the training objective of RL algorithms naturally attenuate parameter updates and avoiding the increase of parameter conflicts as the model converges, favoring a state where *"enough is as good as a feast"*. In contrast, SFT applies update with fixed intensity regardless of convergence. This intrinsic characteristics of RL alleviates *task conflict* during integration. (3) **The joint optimization over positive and negative samples**: RL simultaneously optimizes over both positive and negative examples, whereas SFT relies only on positive samples. This leads to more unbiased updates, resulting in higher convergence stability and more robust task integration.

In summary, this work highlights a previously underappreciated advantage of RL: its natural superiority for model merging. Our findings offer new insights into how different fine-tuning paradigms shape the task conflicts of LLMs and point toward more robust and scalable strategies for building generalist models without retraining from scratch.

**Our key contributions are as follows:**

- We investigate the effect of LLM post-training paradigms on model merging. Our empirical results consistently demonstrate that RL-trained models are more suitable for merging, irrespective of the employed merging methods, the selection of RL algorithms, or the choice

of base models. Further analysis suggests that the superiority of RL-trained models stems from its ability to mitigate task conflicts.

- Through both empirical and theoretical analyses, we advance novel hypotheses and argue that (1) the use of on-policy data, (2) the intrinsic characteristics of RL algorithms, and (3) the joint optimization of positive and negative samples collectively contribute to the enhanced suitability of RL-trained models for model merging.

## 2 BACKGROUND

### 2.1 MODEL MERGING

Model merging aims to integrate the multiple independently trained models into a single unified model. Assume there are $T$ models with parameters $\boldsymbol{\theta}_1, \boldsymbol{\theta}_2, \cdots, \boldsymbol{\theta}_T$, each sharing the same architecture and initialized from a common base model $\boldsymbol{\theta}_0$. The goal of merging is to obtain a new model $\boldsymbol{\theta}_{\text{merge}}$ by applying a merging operator:

$$\boldsymbol{\theta}_{\text{merge}} = \text{merge}(\boldsymbol{\theta}_1, \boldsymbol{\theta}_2, \cdots, \boldsymbol{\theta}_T). \tag{1}$$

Recent advances merging methods are typically based on the task-relevant parameter updates $\boldsymbol{\tau}_i := \boldsymbol{\theta}_i - \boldsymbol{\theta}_0$, namely *Task Vector* (Ilharco et al., 2023). The direction of these *Task Vectors* often reveals conflicts between different tasks. To mitigate such conflicts, recent studies have introduced pruning processes to remove redundant or unimportant parameters, as seen in approaches like DARE (Yadav et al., 2023; Yu et al., 2024; Wang et al., 2024).

### 2.2 SUPERVISED FINE-TUNING AND REINFORCE LEARNING

**Supervised Fine-Tuning (SFT)** Given a dataset $\mathcal{D}_{\text{SFT}} = \{(x_i, y_i)\}_{i=1}^N$, consisting of prompts $x_i$ and their ground-truth responses $y_i$, SFT optimizes the model by minimizing the negative conditional likelihood of generating the correct response:

$$\mathcal{L}_{\text{SFT}}(\boldsymbol{\theta}) = -\mathbb{E}_{(x,y) \sim D_{\text{SFT}}} \left[ \sum_{t=1}^{|y|} \log \pi_\theta(y_t \mid x, y_{<t}) \right], \tag{2}$$

where $\pi_\theta$ denotes the model policy parameterized by $\boldsymbol{\theta}$. This objective enforces strict adherence to labeled data, enabling stable convergence but limiting the flexibility to incorporate reward-based feedback beyond supervised signals.

**Reinforce Learning (RL)** provides an alternative optimization paradigm that aligns model behavior with task-specific reward functions. Formally, the objective is to maximize the expected return:

$$\mathcal{J}_{\text{RL}}(\boldsymbol{\theta}) = \mathbb{E}_{x \sim \mathcal{D}_{\text{RL}}, \, y \sim \pi_\theta(\cdot|x)} \left[ r(y, x) \right], \tag{3}$$

where $r(y, x)$ is a scalar function that reflects the expected reward of the output $y$ given the input $x$. In the era of large language models, a widely used reinforcement learning algorithm is Proximal Policy Optimization (PPO, Schulman et al. (2017)), which can be formulated as:

$$\mathcal{J}^{\text{PPO}}(\boldsymbol{\theta}) = \mathbb{E}_{x \sim \mathcal{D}_{RL}} \sum_{t=1}^{|y|} \left[ \min \left( \frac{\pi_\theta(y_t|y_{<t})}{\pi_{\theta_{\text{old}}}(y_t|y_{<t})} \hat{A}_t, \text{clip} \left( \frac{\pi_\theta(y_t|y_{<t})}{\pi_{\theta_{\text{old}}}(y_t|y_{<t})}, 1 - \epsilon, 1 + \epsilon \right) \hat{A}_t \right) \right] \tag{4}$$

where $\hat{A}_t$ denotes the estimated advantage based on the reward $r$ and the function $\text{clip}(\cdot)$ constrains the probability ratio to the interval $[1 - \epsilon, 1 + \epsilon]$ to ensure training stability. Standard PPO typically employs a critic model to estimate the advantage $A_t$. To further improve the training efficiency, recent studies have proposed critic-free variants, such GRPO (Shao et al., 2024), and REINFORCE++ (Ahmadian et al., 2024), which are also widely adopted in current research.

## 3 REINFORCEMENT LEARNING MITIGATES TASK CONFLICTS

In this section, we experimentally compare the performance preservation of SFT-trained and RL-trained models after model merging.

|  | Math | Code | IF | Puzzle | Rank | Average |
|---|---|---|---|---|---|---|
| SFT | 61.9 | 60.5 | 63.9 | 86.2 | 52.8 | 61.5 |
| Averaging | 52.0(-16%) | 56.0(-7.4%) | 49.2(-23%) | 30.8(-65%) | 51.6(-2.3%) | 47.9(-22%) |
| TIEs | 56.8(-8.3%) | 58.0(-4.1%) | 47.5(-25%) | 35.8(-58%) | 51.3(-2.7%) | 49.9(-19%) |
| Arithmetic | 52.4(-15%) | 56.3(-7.0%) | 48.2(-25%) | 44.9(-48%) | 51.1(-3.3%) | 50.6(-18%) |
| DARE | 58.2(-6.1%) | 58.0(-4.1%) | 46.7(-27%) | 38.0(-56%) | 49.3(-6.7%) | 50.0(-19%) |
| RL (GRPO) | 64.6 | 65.6 | 90.0 | 85.2 | 55.7 | 72.2 |
| Averaging | 62.1(-3.9%) | 61.7(-5.9%) | 84.4(-6.2%) | 37.8(-56%) | 54.4(-2.3%) | 60.1(-17%) |
| TIEs | 63.3(-2.0%) | 64.3(-2.0%) | 90.0(-0%) | 64.6(-24%) | 53.1(-4.7%) | 67.1(-7.1%) |
| Arithmetic | 62.6(-3.1%) | 63.8(-2.7%) | 89.2(-0.9%) | 60.7(-29%) | 54.6(-2.0%) | 66.2(-8.3%) |
| DARE | 63.5(-1.7%) | 64.2(-2.1%) | 89.9(-0.1%) | 65.0(-24%) | 53.1(-4.7%) | 67.1(-7.1%) |

Table 1: Performance comparison across five tasks using different merging strategies (Averaging, TIEs, Arithmetic and DARE), applied to both SFT and RL (GRPO) models. The values in parentheses indicate the relative performance drop compared to the original unmerged model and less performance drop.

## 3.1 EXPERIMENT SETUP

**Models and Settings** We adopt the open-source models **Llama-3.2-3B** (Grattafiori et al., 2024), **Llama-3.1-8B** (Grattafiori et al., 2024), and **Mistral-Small-3-24B**[2] as the base models. For training, we employ three representative RL algorithms: PPO (Schulman et al., 2017), GRPO (Shao et al., 2024), and REINFORCE++ (Ahmadian et al., 2024). Unless otherwise specified, all experiments and analyses default to **Llama-3.1-8B** as the base model and **GRPO** as the optimization algorithm.

**Experiment Tasks and Data** Our evaluation spans five tasks that permit automatic verification: **math**, **code**, **instruction following (IF)**, **logical puzzles (puzzle)**, and **ranking (rank)**. For **math**, models are trained on a subset of *OpenMathInstruct-2* (Ahmad et al., 2025) and evaluated on GSM8K (Cobbe et al., 2021) and MATH-500 (Hendrycks et al., 2021; Lightman et al., 2023). For **code**, models are trained on a subset of *OpenCodeInstruct* (Ahmad et al., 2025) and evaluated on the HUMANEVAL (Chen et al., 2021) and MBPP (Austin et al., 2021) datasets. For **instruction following**, models are trained on the instruction subset from *Tulu-3-SFT* (Lambert et al., 2025) and evaluated on IFEVAL (Zhou et al., 2023b) and the instruction-following subset of LIVEBENCH (White et al., 2024). For **logical puzzles**, models are trained on task *Knights and Knaves* (Johnson-Laird & Byrne, 1990) with synthetic data using templates implemented by Xie et al. (2024) and evaluated on the same task. For **rank**, models are trained on the *Rank1* dataset (Weller et al., 2025) and evaluated on the pairwise ranking benchmark NEVIR. Further details are listed in appendix E.

**Model Merging Settings** We evaluate four model merging strategies: *model averaging* (*Averaging* for short) (Choshen et al., 2022), *TIEs* (Yadav et al., 2023), *Task-Arithmetic* (*Arithmetic* for short) (Ilharco et al., 2023) and *DARE+TIEs* (*DARE* for short) (Yu et al., 2024). In our experiments, we focus on *pairwise merging*, where two models are merged at a time to investigate the robustness and compatibility of different training paradigms. For each task, we report the average performance of the pairwise-merged models from the specific task to any other tasks.

## 3.2 MAIN RESULTS

Table 1 compares the effects of four parameter-merging methods on models trained via SFT and GRPO. A consistent, paradigm-dependent gap emerges: **RL-trained models are substantially more suitable for merging**, largely preserving the performance of individual models after merging, both in specific tasks and in overall averages. SFT-trained models suffer severe degradation—for instance, Puzzle drops by up to 65% and the mean decline spans 18–22%. By contrast, RL-trained models limit losses to under 10% for most strategies. Even on the most fragile task (Puzzle), RL-based models degrade less sharply. This advantage holds across methods: with *Averaging*, SFT shows a 22% mean decline versus 17% for RL; with *TIEs*, RL falls only 7.1% compared to 19% for

---

[2]https://mistral.ai/news/mistral-small-3

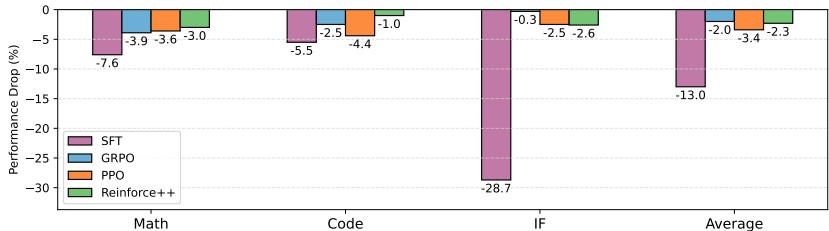

Figure 2: Performance comparison across three tasks using TIEs merging, applied to both SFT and RL models. The values in parentheses indicate the relative performance drop compared to the original unmerged model.

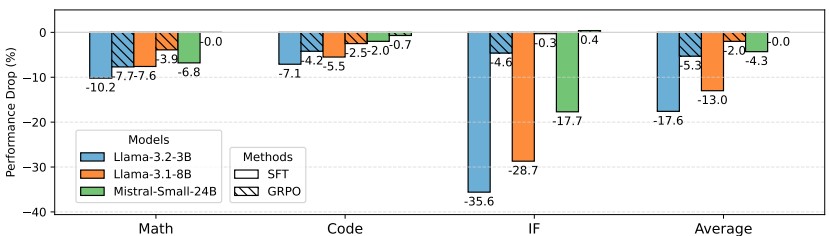

Figure 3: Performance comparison across three tasks using TIEs merging, applied to both SFT and GRPO models with different base models. The values in parentheses indicate the relative performance drop compared to the original unmerged model.

SFT. *Arithmetic* and *DARE* follow the same pattern, with RL consistently outperforming SFT by a wide margin.

**Generality of RL Algorithms in Model Merging**  To further assess the generality of RL algorithms in the context of model merging, we extend our analysis beyond GRPO by incorporating two additional algorithm-PPO and REINFORCE++-with TIEs merging. As shown in Figure 2, across all RL algorithms, the merged models consistently exhibit significantly lower performance degradation compared to those trained with SFT. After TIEs merging, SFT models exhibit a substantial 28.7% decrease in IF, whereas RL models experience only negligible performance losses (GRPO: -0.3%, PPO: -2.5%, REINFORCE++: -2.6%). This reinforces our key observation: **RL-trained model bases are substantially more suitable for model merging than their SFT counterparts**, regardless of the specific RL algorithm employed. Detailed results with other merging methods are provided in Table 4 in the Appendix.

**RL Algorithms Benefit Model Merging with Different LLMs**  We further conduct experiments on different base models, Llama-3.2-3B, Llama-3.1-8B and Mistral-Small-3-24B, across three tasks: Math, Instruction Following, and Code. We evaluate merged models TIEs merging. As shown in Figure 3, across all base models, GRPO-trained models consistently suffer substantially less performance degradation than their SFT-trained counterparts. For instance, after TIEs merging, SFT models drop by -35.6% to -17.7% on IF, whereas RL-trained models decline -4.6% to +0.4%. These results reinforce our central observation: **RL-trained models are substantially more suitable for model merging**, irrespective of the base models. Detailed results with other merging methods are included in Table 5 in the appendix.

### 3.3   PERFORMANCE LANDSCAPE OF SFT- AND RL-TRAINED MODELS

To understand **why RL-trained models are more suitable for model merging**, we investigate the source of this superiority from two aspects: "RL-trained models are more robust to parameter permutation" and "RL mitigates the task conflict". Specifically, given two models fine-tuned independently on tasks $t_1$ and $t_2$, let their respective parameters be denoted as $\boldsymbol{\theta}_{t_1}$ and $\boldsymbol{\theta}_{t_2}$. Model merging

aims to obtain a unified parameter set $\boldsymbol{\theta}_{\mathrm{merge}}$ that integrates knowledge from both tasks. From the perspective of model $\boldsymbol{\theta}_{t_1}$, the merged model can be expressed as a perturbation in parameter space:

$$\boldsymbol{\theta}_{\mathrm{merge}} = \boldsymbol{\theta}_{t_1} + \Delta\boldsymbol{\theta}, \tag{5}$$

where $\Delta\boldsymbol{\theta} = \boldsymbol{\theta}_{\mathrm{merge}} - \boldsymbol{\theta}_{t_1}$ represents the update direction induced by merging with $\boldsymbol{\theta}_{t_2}$. This formulation naturally raises the question: *Is the performance loss of the model caused by simple parameter perturbations, or by parameters that conflict with other tasks $\Delta\boldsymbol{\theta}$?*

To investigate this, we visualize the performance landscape (Li et al., 2018) around $\boldsymbol{\theta}_{t_1}$. For each model, we compare two perturbation directions: (1) Task-induced direction: $\Delta\boldsymbol{\theta} = \boldsymbol{\theta}_{\mathrm{merge}} - \boldsymbol{\theta}_{t_1}$, (2) Random direction: $\boldsymbol{\theta}_{\mathrm{rand}} \sim \mathcal{N}(0, \sigma^2 I)$, scaled such that $\|\boldsymbol{\theta}_{\mathrm{rand}}\|_2 = \|\Delta\boldsymbol{\theta}\|_2$. We evaluate model performance over the two-dimensional surface defined by:

$$f(\alpha, \beta) = \mathcal{L}(\boldsymbol{\theta}_{t_1} + \alpha\Delta\boldsymbol{\theta} + \beta\boldsymbol{\theta}_{\mathrm{rand}}), \tag{6}$$

where $\mathcal{L}(\cdot)$ denotes the task-specific performance function, and $(\alpha, \beta) \in \mathbb{R}^2$ parameterize the perturbation magnitude along each direction.

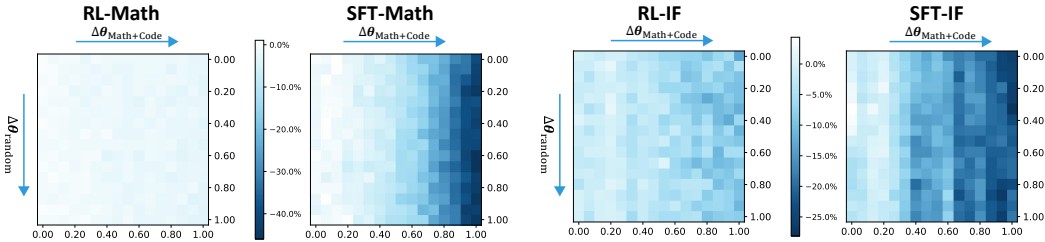

Figure 4: Performance landscape visualization around $\boldsymbol{\theta}_{t_1}$ for SFT and RL models. The $\alpha$-axis corresponds to the task-induced update $\Delta\boldsymbol{\theta}$, and the $\beta$-axis corresponds to a random perturbation of equal norm.

As illustrated in Figure 4[3], adding a random perturbation $\boldsymbol{\theta}_{\mathrm{rand}}$ has minimal impact on performance in both SFT and RL models, indicating robustness to parameter noise. However, the task-induced update $\Delta\boldsymbol{\theta}$ reveals a stark contrast: for SFT-trained models, increasing $\alpha$ along $\Delta\boldsymbol{\theta}$ leads to a noticeable degradation in task performance, suggesting strong interference from task $t_2$. In contrast, RL-trained models maintain stable performance even when perturbed in the direction of $\Delta\boldsymbol{\theta}$, implying that parameters learned from task $t_2$ do not conflict with task $t_1$. These observations support our key finding: **parameter updates induced by SFT tend to be more entangled across tasks, leading to interference, while RL encourages task-orthogonal updates that are less disruptive when merged**.

## 4 HOW RL MITIGATES THE TASK CONFLICT

Having established that RL-trained models exhibit significantly lower task interference than those trained with SFT, we now delve deeper into the mechanisms underlying this advantage. In this section, we systematically analyze how RL can mitigate task conflicts by examining three key differences between RL and SFT, which specifically are on-policy data, different training objective and joint use of positive and negative examples.

### 4.1 THE EFFECT OF ON-POLICY DATA

One main difference between SFT and RL is the training data. In RL, the training data are sampled from the current model (on-policy) while in SFT, it is from a fixed dataset (off-policy). To quantify this effect, we compute the norm of the whole model updates under the three training paradigm: SFT, RL and Rejection-Fine Tuning (RFT, **?**). Notably, RFT can be viewed as a special case of SFT where the training data are sampled from the original model. Table 2 shows that on-policy data consistently produce significantly lower parameter magnitudes than off-policy SFT. For instance, in

---

[3]Full content is shown in appendix D.1.

math task, the norm fo models trained with SFT is 6.5, much higher than that of RFT- and RL-trained models, which are 2.36 and 0.78. These results demonstrate that on-policy data help to regulate the magnitude of gradient updates and reduce parameter sensitivity.

A closer examination reveals that the reduced norm of RL and SFT updates stems from a broader distribution of low-magnitude parameter changes. Considering the proportion of parameters with update magnitudes exceeding $10^{-5}$, RL exhibits only 25.0%, 20.7%, and 24.1% for the math, code, and IF tasks, respectively, in stark contrast to the substantially higher proportions of 79.9%, 78.0%, and 73.9% observed under SFT. This pattern is consistent with recent findings (Mukherjee et al., 2025), which suggests that the dominance of low-magnitude updates in RL-trained models helps preserve the knowledge acquired from other tasks. Therefore, the use of on-policy data reduces the likelihood of parameter conflicts and makes RL-trained models more suitable for model merging.

## 4.2 THE INTRINSIC CHARACTERISTICS OF REINFORCEMENT LEARNING

Another difference of SFT and RL is the optimization targets. Through theoretical and empirical analyses, we find that different optimization targets leading to a natural property of RL, which is "RL optimization is inherently adaptive". Specifically, it adjusts updates according to the model's current capacity and performance. As training progresses, this adaptivity reduces the effective update magnitude, which in turn moderates parameter sensitivity and lowers the likelihood of harmful cross-task interference.

**Theorem 1.** For a single query $x \in \mathcal{D}_{\text{RL}}$, the expected absolute advantage is upper bounded by

$$\mathbb{E}_{a \sim \pi\theta(\cdot|x)}\big[|A(a,x)|\big] \leq \sqrt{\text{Var}(r(a,x))}, \tag{7}$$

where $A(a,x) := r(a,x) - b(x)$ denotes the advantage, computed as the deviation of the observed reward $r(a,x)$ from a baseline $b(x)$.

**Theorem 2.** Based on theorem 1, the advantage estimate $A_n(a,x)$ at the $n$-th step converges to zero in expectation. Since rewards are bounded within a fixed interval and advantages have zero mean for each state, we obtain

$$\lim_{n \to \infty} \mathbb{E}(|A_n(a,x)|) = 0. \tag{8}$$

Theorem 2 implies that as training stabilizes, the expected scaling factor $A$ diminishes. Consequently, RL progressively down-weights parameter updates, whereas SFT continues to apply updates of similar magnitude even at convergence. We list the proof in the appendix B.1. For recently widely used "normalized advantage", $A(a,x) := \frac{r(a,x) - \mathbb{E}(r)}{std(r)}$, the theorem 2 also works. We also list the proof in the appendix B.2.

**Update Magnitudes.** For $n$ sampled trajectories $\{s\}_{s=1}^n$, the cumulative parameter update magnitudes can be expressed as

$$\big\|\Delta\boldsymbol{\theta}_{t_i}^{\text{SFT}}\big\|_2 = \Big\|\sum_{s=1}^n \eta\, \boldsymbol{G}_{t_i}^s\Big\|_2, \qquad \big\|\Delta\boldsymbol{\theta}_{t_i}^{\text{RL}}\big\|_2 = \Big\|\sum_{s=1}^n \eta\, A_{t_i}^s \boldsymbol{G}_{t_i}^s\Big\|_2 \leq \eta \sum_{s=1}^n \Big|A_{t_i}^s\Big|\Big\|\boldsymbol{G}_{t_i}^s\Big\|,$$

where $\boldsymbol{G}_{t_i}^s = \nabla_\theta \log \pi_\theta(y^s|x^s)$ denotes the stochastic gradient contribution of sample $s$. The key difference is that RL scales each update by its corresponding advantage $A_{t_i}^s$, which decays over training (Theorem 2), while SFT applies uniform updates. Similar to the analysis in §4.1, the less updates could reduce the influence of conflicted parameter.

**Conflict norm** One way to understand how RL reduces task interference is to measure the consistency of parameter updates across tasks. Task conflicts occur when two tasks push parameters in opposing directions, leading to destructive interference. To capture this effect, we define the **conflict indicator matrix** as

$$\mathcal{C}(\Delta\boldsymbol{\theta}_{t_i}, \Delta\boldsymbol{\theta}_{t_j}) = \Delta\boldsymbol{\theta}_{t_i} \odot \Delta\boldsymbol{\theta}_{t_j} \tag{9}$$

where $\odot$ denotes the Hadamard (element-wise) product. Negative entries of $\mathcal{C}$ indicate conflicting updates, while positive entries correspond to aligned updates. We then define the **conflict norm** as:

$$\big\|\mathcal{C}(\Delta\boldsymbol{\theta}_{t_i}, \Delta\boldsymbol{\theta}_{t_j})\big\|_{\text{conflict}} = \Big\|\big(\mathcal{C}(\Delta\boldsymbol{\theta}_{t_i}, \Delta\boldsymbol{\theta}_{t_j})_{ij} \cdot \mathbf{1}_{\{\mathcal{C}_{ij}<0\}}\big)_{i,j}\Big\|_2 \tag{10}$$

|      | **Math** | **Code** | **IF** |
|------|----------|----------|--------|
| SFT  | 6.50     | 7.75     | 4.83   |
| RFT  | 2.36     | 2.17     | 1.70   |
| RL   | **0.78** | **0.71** | **0.64** |

Table 2: The norm of $\Delta\boldsymbol{\theta}$ for different tasks and settings.

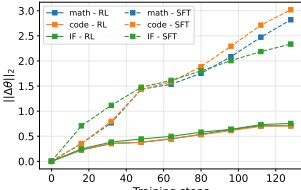

Figure 5: The norm of SFT and RL

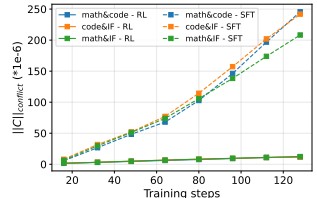

Figure 6: The conflict norm of SFT and RL

where $\mathbf{1}_{\{\mathcal{C}_{ij}<0\}}$ is an indicator function that selects only the conflicting entries. This measure thus quantifies the aggregate strength of parameter conflicts between two tasks.

**Implications for Task Interference.** Building on these results, we examine the conflict norm between two tasks $t_1$ and $t_2$. Under SFT:

$$\|\mathcal{C}(\Delta\boldsymbol{\theta}_{t_1}, \Delta\boldsymbol{\theta}_{t_2})\|_{\text{conflict}}^{\text{SFT}} = \eta^2 \|\mathcal{C}(\sum_{k=1}^{n} \boldsymbol{G}_{t_1}^k, \sum_{k=1}^{n} \boldsymbol{G}_{t_2}^k)\|_{\text{conflict}},$$

whereas under RL:

$$\|\mathcal{C}(\Delta\boldsymbol{\theta}_{t_1}, \Delta\boldsymbol{\theta}_{t_2})\|_{\text{conflict}}^{\text{RL}} = \eta^2 \|\mathcal{C}(\sum_{k=1}^{n} A_1^k \boldsymbol{G}_{t_1}^k, \sum_{k=1}^{n} A_2^k \boldsymbol{G}_{t_2}^k)\|_{\text{conflict}}.$$

In the verifiable setting where $r \in \{0, 1\}$, we have $|A| \leq \sqrt{\text{Var}(r)} \leq \frac{1}{2}$, and $A^k \to 0$ in expectation as training stabilizes. It follows that

$$\mathbb{E}\left[\|\mathcal{C}(\Delta\boldsymbol{\theta}_{t_1}, \Delta\boldsymbol{\theta}_{t_2})\|_{\text{conflict}}^{\text{RL}}\right] \ll \mathbb{E}\left[\|\mathcal{C}(\Delta\boldsymbol{\theta}_{t_1}, \Delta\boldsymbol{\theta}_{t_2})\|_{\text{conflict}}^{\text{SFT}}\right] \tag{11}$$

which formalizes the intuition that RL reduces cross-task parameter conflicts by down-scaling gradient magnitudes through the vanishing advantage.

We plot the norm of update parameter in Figure 5 and the conflict norm in Figure 6. As is shown in the figures, the growth trend of norm and conflict norm of RL is obviously slower than SFT, indicating that the advantages effect during the training process.

### 4.3 ANALYSIS OF OPTIMIZATION OVER BOTH POSITIVE AND NEGATIVE SAMPLES

The third difference between RL and SFT is that RL is optimized over both positive and negative samples. In this section, we provide empirical analyses to elucidate the effects of optimization over both positive and negative samples.

To further isolate the contribution of negative samples, we design a controlled experiment in which their influence is selectively removed. Specifically, we construct an RL variant (**RL-Pos**) in which the advantage values for all negative samples are set to zero, thereby excluding them from gradient updates while retaining the KL regularization and on-policy sampling.

We test two hypotheses: **H1 (Single-task performance).** Models trained with both positive and negative samples should converge to higher task-specific accuracy than models trained on positive samples only. **H2 (Cross-task conflict).** Given the same training budget, models trained with both types of samples should exhibit lower cross-task conflict (as measured by performance degradation after model merging) compared to positive-only training.

To validate **H1**, we report the convergent accuracy of three models—SFT, RL, and RL-Pos—on Math, Code, and IF tasks in Table 3. Results show that while RL-Pos improves over SFT, it still underperforms full RL, confirming that negative samples facilitate better single-task optimization. To test **H2**, we select RL and RL-Pos checkpoints trained for the same number of steps and apply two merging strategies—parameter averaging and Task-Independent Experts (TIEs). As shown in Figure 7, RL consistently suffers less performance degradation, reinforcing the claim that jointly optimizing both positive and negative examples in RL guides the model toward an unbiased, task-specific subspace and further reduces parameter conflicts.

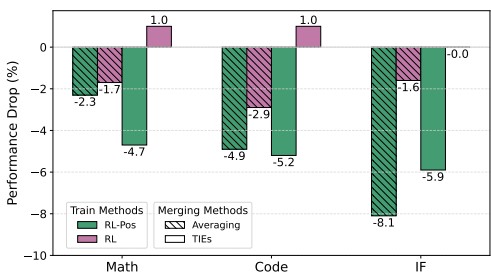

|  | **Math** | **Code** | **IF** | **Avg.** |
|---|---|---|---|---|
| SFT | 61.9 | 60.5 | 63.9 | 62.1 |
| RL | **64.6** | **65.6** | **90.0** | **73.4** |
| RL-Pos | 58.5 | 61.7 | 86.1 | 68.8 |

Table 3: Convergent performance across tasks.

Figure 7: Performance drop in cross-task merging for RL and RL-Pos models under model averaging and TIEs.

## 5 RELATED WORKS

**Model Merging of Large Language Models** Model merging offers a practical alternative to retraining by eliminating the need for access to raw training data or computationally expensive fine-tuning procedures (Ilharco et al., 2022; Crisostomi et al., 2024; Yang et al., 2024a). In this work, we focus on training-free model merging, where the merging process relies solely on the weights of pre-trained and fine-tuned models. Recent advances have improved this approach through heuristically guided merging strategies based on parameter statistics (e.g. value magnitudes or sign alignment) (Yadav et al., 2023), task importance derived from the Fisher Information Matrix (Tam et al., 2023), task-specific layer-wise attribution modeling (Wang et al., 2024), and trust region constraints (Sun et al., 2025). Several efforts have also aimed to explain the effectiveness of model merging from a theoretical perspective, drawing on tools such as loss landscape geometry (Izmailov et al., 2018; Gupta et al., 2020), bias-variance decomposition (Arpit et al., 2022; Rame et al., 2022), and linear mode connectivity (LMC) in neural networks (Frankle et al., 2020; Zhou et al., 2024). However, these analyses have been predominantly confined to models fine-tuned via SFT. To date, little attention has been paid to how different fine-tuning paradigms—particularly reinforcement-based methods—affect the model merging and interaction of models.

**Analysis of LLM Post-Training** Post-training has emerged as an effective approach for enhancing the task-specific capabilities of large language models (LLMs) (Grattafiori et al., 2024; Team, 2024; Shao et al., 2024). Among the most commonly used paradigms, **SFT** adapts pre-trained models to downstream tasks by training them on task-specific datasets, often formatted as instructions (Wei et al., 2021; Zhou et al., 2023a; Chung et al., 2024; Li et al., 2024). In contrast, **Reinforcement Learning (RL)** is typically employed to align models with human preferences or to optimize performance on specific target tasks (Ouyang et al., 2022; Ahmadian et al., 2024; Guo et al., 2025; Yang et al., 2025; Zhao et al., 2025). To deepen understanding of the post-training stage, recent research has explored how different learning paradigms influence model behavior. For instance, several studies investigate memorization and generalization dynamics across knowledge-intensive and reasoning tasks (Allen-Zhu & Li, 2023; Ye et al., 2024; Qi et al., 2024; Chu et al., 2025; Kang et al., 2025). Others have examined learning dynamics by contrasting SFT and RL in terms of convergence behavior and sample efficiency (Ren & Sutherland, 2024; Zeng et al., 2025; Kang et al., 2025). Recent research has also highlighted distinctive properties of RL compared with SFT, showing that RL tends to produce sparser parameter updates (Shenfeld et al., 2025) and can help mitigate catastrophic forgetting (Mukherjee et al., 2025). In this work, we take a complementary perspective by analyzing how the choice of post-training paradigm impacts *task conflicts* in model merging. Our findings demonstrate that models fine-tuned with RL exhibit significantly reduced inter-task interference compared to those trained with SFT, thereby offering a more robust foundation for multi-task integration.

## 6 CONCLUSION

This work provides a comprehensive investigation into the influence of post-training paradigms on model merging in LLMs. Our central finding is that RL, as opposed to standard SFT, inherently

mitigates cross task conflicts, making it more substantially suitable for model merging. To further understand the mechanism behind this advantage, we isolate and evaluate three components of the RL training objective: (1) on-policy training data, (2) the intrinsic characteristics of RL algorithms, and (3) optimization over both positive and negative samples. Both theoretical and empirical analyses demonstrate that the three components play a critical role in conflict mitigation. Taken together, our findings indicate that RL is not merely an alternative to SFT but constitutes a fundamentally more suitable paradigm for multi-task post-training in foundation models. Beyond this, the results provide new insights into how different fine-tuning paradigms shape task conflicts in LLMs and highlight RL as a robust and scalable strategy for developing generalist models.

## ETHICS STATEMENT

All authors have read and comply with the ICLR Code of Ethics. This study presents an analysis of the impact of different post-training paradigms—specifically, SFT and RL—on the model merging of LLMs. All experimental components, including models, datasets, and benchmarks, are drawn from publicly available sources and are well-established in the academic literature. Consequently, the research employs no proprietary or sensitive data, involves no human subjects, and presents no additional foreseeable risks beyond those associated with the broader field of LLM research.

## REPRODUCIBILITY STATEMENT

Our experiments are built upon open-source models, datasets, and algorithms for both RL training and model merging. We employ greedy decoding for all evaluations. Note that minor variations in results may occur due to hardware differences or inference framework implementations.

## ACKNOWLEDGMENT

The research work has been supported by the Natural Science Foundation of China under Grant No. 62476271

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

## A THE USE OF LARGE LANGUAGE MODELS

In this study, LLMs are employed solely to assist with the refinement of written expression. Specifically, they were used to improve grammar, enhance clarity, and ensure fluency in academic writing. The LLMs were not involved in data analysis, methodological design, or interpretation of results, but served only as a linguistic polishing tool to improve readability and presentation.

## B PROOF OF THE CONVERGENCE OF $\mathbb{E}(A_n)$

### B.1 UPPER BOUND ESTIMATION FOR ADVANTAGE WITH BASELINE

In the standard formulation of advantage estimation with a baseline $\hat{b}$ (Williams, 1992),

$$A := r - \hat{b},$$

where $\hat{b} = \mathbb{E}[r_n]$ is the expectation of the reward $r$. Then we obtain the following bound:

$$
\begin{aligned}
\mathbb{E}(|A|) &\leq \sqrt{\mathbb{E}(A^2)} \\
&= \sqrt{\mathbb{E}((r - \mathbb{E}(r))^2)} \\
&= \sqrt{\mathbb{E}(r^2) - 2\mathbb{E}(r)\mathbb{E}(r) + \mathbb{E}^2(r)} \\
&= \sqrt{\mathbb{E}(r^2) - \mathbb{E}^2(r)} \\
&= \sqrt{\mathrm{Var}(r)}
\end{aligned}
\tag{12}
$$

Recent work has established the global convergence of reinforcement learning algorithms (Zhang et al., 2020). In particular, the expected reward at the $n$-th step, $\mathbb{E}(r_n)$, converges to $r^\star$, the reward of the optimal policy. Formally,

$$\lim_{n \to \infty} \mathbb{E}(r_n) = r^\star. \tag{13}$$

**Variance Decay.** Since the reward is bounded in the interval $[a, r^\star]$, we can apply the Bhatia–Davis inequality:

$$\mathrm{Var}(r_n) = \mathbb{E}(r_n^2) - \mathbb{E}^2(r_n) \leq (r^\star - \mathbb{E}(r_n))(\mathbb{E}(r_n) - a). \tag{14}$$

As $\mathbb{E}(r_n) \to r^\star$, the right-hand side tends to zero, which implies

$$\lim_{n \to \infty} \mathrm{Var}(r_n) = 0,$$

which leads to

$$\lim_{n \to \infty} \mathbb{E}(|A_n|) = 0. \tag{15}$$

### B.2 CONVERGENCE FOR DISCRETE ADVANTAGE WITH BASELINE AND STANDARD DEVIATION

A widely adopted variant of advantage estimation, used for example in GRPO and Reinforce++, normalizes the centered reward by its standard deviation:

$$A := \frac{r - \mathbb{E}(r)}{\sqrt{\mathrm{Var}(r)}}.$$

**Theorem 3.1.** For any bounded reward distribution $P_r$ supported on $[a, b]$, the expectation of the absolute advantage is maximized when $P_r$ is a Bernoulli distribution $P_{r^\star}$ with the same expectation, i.e.,

$$\mathbb{E}_{P_r}(|A|) \leq \mathbb{E}_{P_{r^\star}}(|A|), \qquad \text{with } \mathbb{E}_{P_r}(r) = \mathbb{E}_{P_{r^\star}}(r).$$

*Proof.* Define the centered random variable $X := r - \mu$, where $\mu = \mathbb{E}[r]$. Then the normalized advantage can be written as

$$\mathbb{E}(|A(r)|) = \frac{\mathbb{E}(|X|)}{\sqrt{\mathbb{E}(X^2)}}.$$

Consider the family of reward distributions with fixed expectation:

$$\mathcal{S}_\mu := \{P \mid \mathbb{E}_P(r) = \mu\}.$$

This set is convex and compact under the topology of weak convergence (by Prokhorov's theorem), since all measures are supported on the compact interval $[0, 1]$.

—

**Supporting-Line Reduction.** Define the linear functionals

$$U(P) := \mathbb{E}_P(|X|), \quad V(P) := \mathbb{E}_P(X^2).$$

Maximizing the functional

$$\psi(U, V) := \frac{U}{\sqrt{V}}$$

over $\mathcal{S}_\mu$ can be reduced to maximizing a linear functional. Specifically, if $(u_\star, v_\star)$ is an optimizer of $\psi$, then set

$$c := \frac{u_\star}{\sqrt{v_\star}}, \qquad \lambda := \frac{u_\star}{2 v_\star}.$$

By convex analysis, $\psi$ admits the supporting-line inequality

$$c\sqrt{v} \leq u_\star + \lambda(v - v_\star), \qquad \forall v \geq 0.$$

Hence, for any $(u, v) \in \mathcal{S}_\mu$,

$$u - \lambda v \leq u_\star - \lambda v_\star,$$

which shows that $(u_\star, v_\star)$ also maximizes the linear functional

$$L(u, v) := u - \lambda v$$

over $\mathcal{S}_\mu$. Thus, the nonlinear problem

$$\sup_{(u,v) \in \mathcal{S}_\mu} \frac{u}{\sqrt{v}} \qquad \text{and} \qquad \sup_{(u,v) \in \mathcal{S}_\mu} (u - \lambda v)$$

share the same maximizer $(u_\star, v_\star)$.

—

**Existence of Optimizer.** Since $\mathcal{S}_\mu$ is compact and $L$ is continuous, the supremum is attained at some $(u_\star, v_\star) \in \mathcal{S}_\mu$. By definition, there exists a distribution $P_\star$ such that

$$(u_\star, v_\star) = (U(P_\star), V(P_\star)).$$

Moreover, since $L$ is linear in $(u, v)$ and both $U$ and $V$ are linear functionals of $P$, maximizing $L$ over $\mathcal{S}_\mu$ is equivalent to maximizing

$$P \mapsto \int (|x| - \lambda x^2) P(dx),$$

over the convex, compact set $\mathcal{S}_\mu$. By the Krein–Milman theorem, the supremum is attained at an *extreme point* of $\mathcal{S}_\mu$.

—

**Extreme Points of $\mathcal{S}_\mu$.** The extreme points of $\mathcal{S}_\mu$ are precisely two-point distributions of the form

$$p(x) = c\,\delta(x - x_1) + (1 - c)\,\delta(x - x_2),$$

satisfying the expectation constraint $cx_1 + (1 - c)x_2 = \mu.$, in which $\delta$ is the dirac delta function.

*Proof.*
*Sufficiency (two points ⇒ extreme).* Let $P = c\,\delta_{x_1} + (1 - c)\,\delta_{x_2}$ with $x_1 \neq x_2$ and $c \in (0, 1)$ satisfy $c\,x_1 + (1 - c)\,x_2 = \mu$. Suppose

$$P = \lambda P_1 + (1 - \lambda)P_2, \qquad 0 < \lambda < 1, \quad P_1, P_2 \in \mathcal{S}_\mu.$$

For any Borel set $E \subset \mathbb{R}$ we have $P(E) = \lambda P_1(E) + (1 - \lambda)P_2(E)$. Since $P(\{y\}) = 0$ for every $y \notin \{x_1, x_2\}$, it follows that $P_1(\{y\}) = P_2(\{y\}) = 0$ for all such $y$ (otherwise the convex combination would be positive), hence

$$supp(P_1),\ supp(P_2)\ \subseteq\ \{x_1, x_2\}.$$

Write $P_i = \alpha_i\,\delta_{x_1} + (1 - \alpha_i)\,\delta_{x_2}$ for $i = 1, 2$ with some $\alpha_i \in [0, 1]$. Because $P_i \in \mathcal{S}_\mu$, each must satisfy the same mean constraint:

$$\alpha_i x_1 + (1 - \alpha_i)x_2 = \mu \quad \Rightarrow \quad \alpha_i = \frac{x_2 - \mu}{x_2 - x_1} \quad (i = 1, 2).$$

Thus $\alpha_1 = \alpha_2 = c$, and then the identity $P = \lambda P_1 + (1 - \lambda)P_2$ forces $P_1 = P_2 = P$. Hence $P$ is extreme. The degenerate one–point case $P = \delta_{x^*}$ also yields extremality: the mean constraint implies $x^* = \mu$, and the same support argument shows any convex decomposition must be trivial.

*Necessity ($\geq 3$ points $\Rightarrow$ not extreme).* Suppose the size of $p$'s support, $|supp(p)|$ is greater than 2. Assume $P \in \mathcal{S}_\mu$ has at least three distinct atoms $y_1, y_2, y_3$ with masses $\alpha_1, \alpha_2, \alpha_3 > 0$:

$$P = \sum_{i=1}^{3} \alpha_i\,\delta_{y_i} + P_{\text{rest}}, \qquad \alpha_1 + \alpha_2 + \alpha_3 > 0,$$

.where $P_{\text{rest}}$ is the remainder (possibly zero). Consider the $2 \times 3$ matrix

$$M = \begin{pmatrix} 1 & 1 & 1 \\ y_1 & y_2 & y_3 \end{pmatrix}.$$

Since $\text{rank}(M) = 2$, there exists a nonzero vector $\beta = (\beta_1, \beta_2, \beta_3)^\top$ in the nullspace of $M$, i.e.

$$\beta_1 + \beta_2 + \beta_3 = 0, \qquad \beta_1 y_1 + \beta_2 y_2 + \beta_3 y_3 = 0, \qquad \beta \neq 0.$$

Define the signed measure

$$\nu := \sum_{i=1}^{3} \beta_i\,\delta_{y_i}.$$

Then $\int 1\,d\nu = 0$ and $\int x\,d\nu = 0$. Choose

$$t \in \left(0,\ \min_{i:\,\beta_i \neq 0} \frac{\alpha_i}{|\beta_i|}\right],$$

so that all coefficients $\alpha_i \pm t\beta_i$ remain nonnegative. Set

$$P_1 := P + t\nu, \qquad P_2 := P - t\nu.$$

By construction, $P_1$ and $P_2$ are probability measures ($\int 1\,dP_j = 1$), they satisfy the mean constraint ($\int x\,dP_j = \mu$), and $P = \frac{1}{2}P_1 + \frac{1}{2}P_2$. Since $\nu \neq 0$, we have $P_1 \neq P_2$, showing that $P$ is not extreme.

If $P$ has infinite support or a non-atomic part, pick three disjoint Borel sets of positive mass and perform the same construction after restricting to those sets (approximating each by a point via conditional expectations), which yields a nontrivial $\nu$ with $\int 1\,d\nu = 0$ and $\int x\,d\nu = 0$. Hence any $P$ with $|supp(P)| \geq 3$ fails to be extreme.

Combining the two parts, the extreme points of $\mathcal{S}_\mu$ are precisely the probability measures supported on at most two points, i.e. the two-point laws $c\,\delta_{x_1} + (1 - c)\,\delta_{x_2}$ satisfying $c\,x_1 + (1 - c)\,x_2 = \mu$ (with the one-point Dirac at $\mu$ as a degenerate case).

—

**Convergence of the Discrete Advantage.** Let $b = r*$, from the previous proof, we have

$$\lim_{n \to \infty} E(r_n) = b$$

For the discrete rewards, which lives on a finite grid $\{a, i_1, i_2, \cdots, i_K, b\}$, the extremizer with fixed mean $\mu \in [i_K, b]$ is supported on $\{i_K, b\}$ and

$$\lim_{n \to \infty} \mathbb{E}(|A_n|) = \lim_{\mu \to b} \mathbb{E}\left(\frac{2\sqrt{(b - \mu)(\mu - i_K)}}{b - i_K}\right) = 0. \tag{16}$$

Here $\mu$ is $\mathbb{E}(r)$. This result shows that, under convergence of the RL algorithm, the normalized advantage vanishes asymptotically, ensuring that parameter updates diminish and training stabilizes.

|  | Math | Code | IF | Average |
|---|---|---|---|---|
| SFT | 61.9 | 60.5 | 63.9 | 62.1 |
| Averaging | 55.8(-10%) | 55.7(-7.8%) | 51.7(-19.1%) | 54.4(-12.4%) |
| TIEs | 57.1(-7.6%) | 57.2(-5.5%) | 47.7(-28.7%) | 54.0(-13.0%) |
| GRPO | 64.6 | 65.6 | 90.0 | 73.4 |
| Averaging | 60.9(-5.3%) | 62.0(-5.4%) | 83.7(-7.0%) | 68.9(-6.2%) |
| TIEs | 62.1(-3.9%) | 64.0(-2.5%) | 89.8(-0.3%) | 72.0(-2.0%) |
| PPO | 62.8 | 65.2 | 87.4 | 71.8 |
| Averaging | 60.9(-3.0%) | 61.7(-5.4%) | 80.0(-8.5%) | 67.5(-5.9%) |
| TIEs | 60.5(-3.6%) | 62.3(-4.4%) | 85.2(-2.5%) | 69.3(-3.4%) |
| REINFORCE++ | 62.3 | 63.7 | 83.8 | 70.0 |
| Averaging | 61.5(-1.3%) | 61.0(-4.2%) | 79.1(-5.6%) | 67.2(-4.0%) |
| TIEs | 60.4(-3.0%) | 63.1(-1.0%) | 81.6(-2.6%) | 68.4(-2.3%) |

Table 4: Performance comparison across three tasks using different merging strategies (Averaging and TIEs), applied to both SFT and RL models. The values in parentheses indicate the relative performance drop compared to the original unmerged model.

|  | Math | Code | IF | Average |
|---|---|---|---|---|
| *Based on Llama-3.2-3B* | | | | |
| SFT | 42.7 | 42.8 | 42.2 | 42.6 |
| Averaging | 33.1(-22.5%) | 38.3(-10.4%) | 33.1(-21.6%) | 34.8(-18.2%) |
| TIEs | 38.3(-10.2%) | 39.8(-7.1%) | 27.2(-35.6%) | 35.1(-17.6%) |
| GRPO | 41.4 | 49.8 | 56.7 | 49.3 |
| Averaging | 40.4(-2.4%) | 46.2(-7.2%) | 50.0(-11.8%) | 45.5(-7.6%) |
| TIEs | 38.2(-7.7%) | 47.7(-4.2%) | 54.1(-4.6%) | 46.7(-5.3%) |
| *Based on Llama-3.1-8B* | | | | |
| SFT | 61.9 | 60.5 | 63.9 | 62.1 |
| Averaging | 55.8(-10%) | 55.7(-7.8%) | 51.7(-19.1%) | 54.4(-12.4%) |
| TIEs | 57.1(-7.6%) | 57.2(-5.5%) | 47.7(-28.7%) | 54.0(-13.0%) |
| GRPO | 64.6 | 65.6 | 90.0 | 73.4 |
| Averaging | 60.9(-5.3%) | 62.0(-5.4%) | 83.7(-7.0%) | 68.9(-6.2%) |
| TIEs | 62.1(-3.9%) | 64.0(-2.5%) | 89.8(-0.3%) | 72.0(-2.0%) |
| *Based on Mistral-Small-24B* | | | | |
| SFT | 73.9 | 71.7 | 76.5 | 74.0 |
| Averaging | 71.5(-3.3%) | 71.6(-0%) | 66.5(-13.0%) | 69.9(-5.6%) |
| TIEs | 68.9(-6.8%) | 70.3(-2.0%) | 62.9(-17.7%) | 70.4(-4.3%) |
| GRPO | 77.9 | 73.9 | 89.6 | 80.5 |
| Averaging | 77.0(-1.2%) | 74.0(-0%) | 86.4(-3.6%) | 79.1(-1.7%) |
| TIEs | 77.9(-0%) | 73.4(-0.7%) | 90.0(+0.4%) | 80.4(-0%) |

Table 5: Performance comparison across three tasks using different merging strategies (Averaging and TIEs), applied to both SFT and GRPO models with different base models. The values in parentheses indicate the relative performance drop compared to the original unmerged model.

## C    DETAILED EXPERIMENTS RESULTS

## D    PARAMETER SIGN CONFLICTS

A direct way to assess task interference between large language models (LLMs) is to measure the *parameter sign conflict*, i.e., the proportion of parameters for which the update directions (weight differences) differ between models trained on different tasks. Prior work has shown that parameter sign conflicts can substantially impact merged model performance, as inconsistent signs may lead to destructive interference during parameter fusion (Yadav et al., 2023). To quantify this, we compute the ratio of sign conflicts while varying the *kept parameter rate*, defined as the fraction of parameters

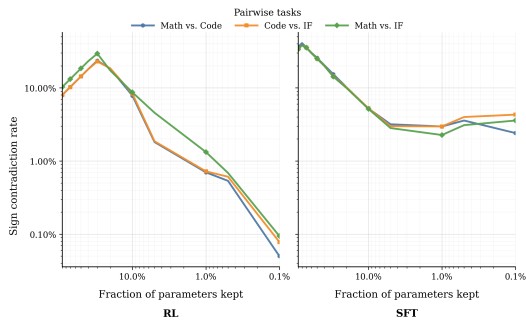

Figure 8: Proportion of parameter sign conflicts under varying proportions of retained parameters.

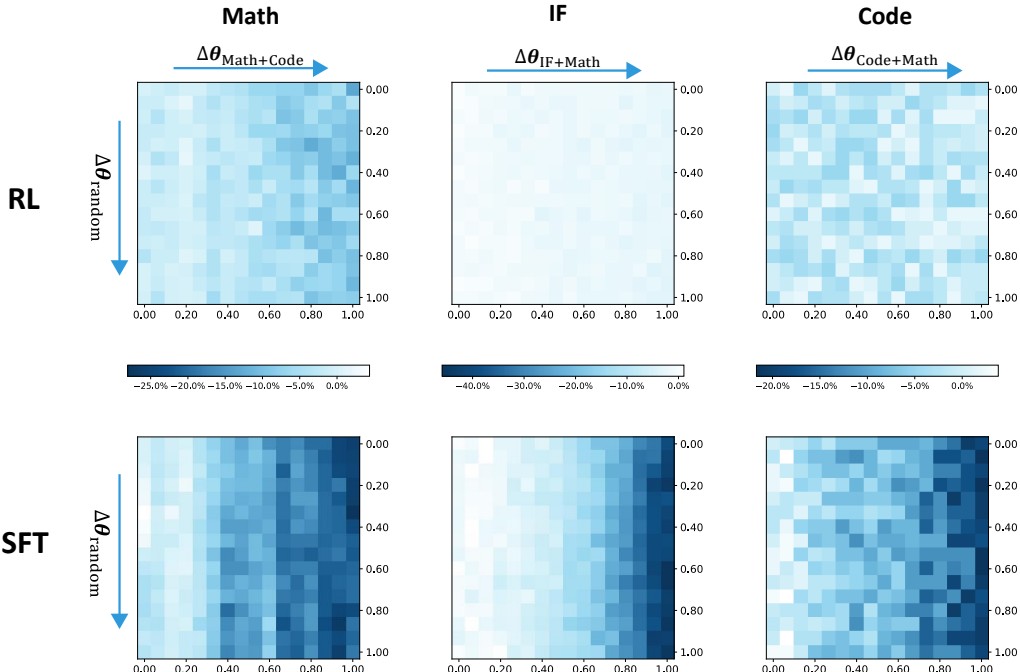

Figure 9: The full content of performance landscape

retained after selecting those with the largest absolute values. Figure 8 reports the conflict rate as a function of the kept rate. As shown, RL-trained models consistently exhibit lower sign conflict rates than their SFT-trained counterparts, particularly in the highest-importance parameter subset. This suggests that RL optimization produces more compatible update directions across tasks, thereby reducing destructive interference during merging.

## D.1 PERFORMANCE LANDSCAPE

## D.2 THE NUMBER OF MERGING MODEL

We examine how the *number of merged models* influences task conflicts under different post-training paradigms. In this experiment, we employ the TIEs merging strategy to combine varying numbers of independently fine-tuned models. As shown in Figure 1, the performance degradation patterns differ markedly between paradigms. For RL-trained models, the average performance decreases only gradually as the number of merged models increases. In contrast, SFT-trained models exhibit a much steeper decline, suggesting that parameter conflicts accumulate more rapidly when merging multiple SFT-based models. This result further supports our earlier findings that RL produces more task-

orthogonal parameter updates, thereby mitigating cross-task interference in multi-model merging scenarios.

# E FUTHER DETAILS

## E.1 EXPERIMENT SETUP DETAILS

**Model and training datasets**  We adopt the open-source model **LLaMA-3.1-8B** (Grattafiori et al., 2024) as the base model for experiments and analysis. Our evaluation covers five tasks that allow for automatic verification: mathematics, code generation, logical puzzles, instruction following, and ranking. For the puzzle task, we adopt the representative logical reasoning benchmark *Knights and Knaves* (Johnson-Laird & Byrne, 1990). For the **SFT** setting, we prepare task-specific training data as follows. For the math task, we use a subset of *OpenMathInstruct-2* (Ahmad et al., 2025). For the code generation task, we adopt a subset of *OpenCodeInstruct* (Ahmad et al., 2025). For the puzzle task, we automatically generate synthetic data using templates implemented by Xie et al. (2024), with the number of people ranging from 2 to 8. For the instruction-following task, we use the instruction subset from *Tulu-3-SFT* (Lambert et al., 2025). For the ranking task, we use the *Rank1* dataset (Weller et al., 2025). For the **RL** setting, we reuse the same query for the math, code, puzzle, and ranking tasks. To enable verifiable supervision for the instruction-following task, we employ the signal construction method proposed by Pyatkin et al. (2025). Before applying reinforcement learning, we first fine-tune the base model on the math, code, and instruction-following datasets to equip it with basic instruction-following capabilities. This initialization step ensures stable and meaningful reward signals during the RL training stage.

**Benchmarks and Evaluation Metrics**  To comprehensively assess model performance across diverse task domains, we employ a broad set of benchmarks. For **mathematical reasoning**, we adopt GSM8K (Cobbe et al., 2021) and MATH-500 (Hendrycks et al., 2021; Lightman et al., 2023), and report the *accuracy* on each benchmark. For **code generation**, we utilize the HUMANEVAL (Chen et al., 2021) and MBPP (Austin et al., 2021) datasets, including both the base and plus versions. The evaluation metric is *pass@1*, which measures the percentage of correct solutions in the first attempt. For **instruction following**, we adopt IFEVAL (Zhou et al., 2023b) and the instruction-following subset of LIVEBENCH (White et al., 2024). We report both the *loose* and *strict* accuracies for IFE-VAL, and the overall *LiveBench score* as computed by the benchmark's official evaluation script. For the **puzzle-solving** task, we generate evaluation data using a templated prompt . Specifically, we construct an *in-domain* test set involving scenarios with 2 to 8 people, and an *out-of-domain (OOD)* test set with 9 to 13 people. Accuracy is reported separately for in-domain and OOD settings. Finally, for **ranking**, we evaluate on the pairwise ranking benchmark NEVIR, and report the *accuracy* computed using the official MTEB evaluation protocol.

## E.2 IMPLEMENTATION DETAILS

For **SFT** experiments, we train models using the OPENRLHF[4] (Hu et al., 2024) framework. Most hyperparameters are adopted from the `tulu-3` configuration, including a learning rate of $5 \times 10^{-6}$, batch size of 128, warm-up ratio of 0.03, a learning rate decay scheduler, and a total of 3 training epochs. For the **Reinforcement Learning (RL)** experiments, we employ the VERL[5] (Sheng et al., 2024) framework and **GRPO** as the RL algorithm. The training configuration includes a learning rate of $1 \times 10^{-6}$, rollout batch size of 512, rollout count of 8, rollout temperature and top-$p$ both set to 1.0, and a KL-divergence coefficient of $1 \times 10^{-3}$. For **model merging**, we utilize the MERGEKIT toolkit. In the case of *TIEs merging* and *DARE*, we follow the default hyperparameter settings recommended by MERGEKIT, setting the sensitivity to 0.8 and the interpolation weight to 0.5. In *Task-Arithmetic*, we set $\lambda = 0.7$, which is recomended by the original paper.

---

[4] https://github.com/OpenRLHF/OpenRLHF
[5] https://github.com/volcengine/verl

