# OpenReview forum: "Enough is as good as a feast: A Comprehensive Analysis of How Reinforcement Learning Mitigates Task Conflicts in LLMs"
_ICLR.cc/2026/Conference — ICLR 2026 Poster_

### Official Review · Reviewer_W4Vs · 2025-10-25

**Soundness:** 2
**Presentation:** 3
**Contribution:** 3
**Rating:** 6
**Confidence:** 3

**Summary:**

This paper presents a systematic empirical and theoretical analysis of how reinforcement learning (RL) and supervised fine-tuning (SFT) affect model merging performance in large language models (LLMs). By evaluating five representative NLP and reasoning tasks, the authors demonstrate and dissect why RL-trained models are more robust to parameter/task conflicts and preserve higher performance post-merging. The study identifies three contributing mechanisms: reduced parameter update magnitude due to on-policy data in RL, diminishing update strength from RL’s adaptive optimization objectives, and the role of joint positive/negative example optimization in RL. These analyses collectively suggest RL-trained LLMs are better-suited for consolidation via model merging than their SFT-trained counterparts.

**Strengths:**

1.	Comprehensive Empirical study: This paper systematically explores the effects of SFT vs. RL across multiple LLMs, tasks (math, code, instruction following, logical puzzles, ranking), and merging algorithms (Averaging, TIEs, Arithmetic, DARE), providing robustness to the experimental findings (see Table 1, Table 4, Table 5).
2.	The writing is meticulous: The paper proposes three hypothesis of why RL performs better than SFT from three different perspectives and verify these hypothesis both from theoretical analysis and experiments

**Weaknesses:**

1.	The experiments focus on pairwise merges, but the field is also interested in merging larger ensembles of specialized models. Do the observed RL advantages scale when merging 3, 4, or more models across highly distinct or hierarchical tasks? If not, are there any limiting factors?
2.	For some tasks/methods, performance loss is reported as -1% to -4%. Are these differences statistically significant? Including error bars or statistical significance tests would add clarity and reliability to interpretation.

**Questions:**

See weaknesses.

---

> ### Author Response · Authors · 2025-11-21
>
> ## Question1: The experiments focus on pairwise merges, but the field is also interested in merging larger ensembles of specialized models. Do the observed RL advantages scale when merging 3, 4, or more models across highly distinct or hierarchical tasks? If not, are there any limiting factors?
>
>
> We appreciate the reviewer’s insightful question regarding the scalability of RL-induced advantages when merging larger ensembles of specialized models. In addition to the 2-model results presented in the paper, we conducted extended experiments merging **3, 4, and 5 specialized models**.
>
> THe results are illustrated in Figure 1 and detailed in the accompanying tables. Across all tasks (Math, Code, IF, Puzzle, Rank), the results consistently show that **RL-trained models exhibit substantially smaller degradation than SFT-trained models as the number of merged models increases**. For example, averaged over all tasks, SFT performance drops sharply from –19% (2 models) to –69% (5 models), whereas RL degradation remains controlled, from –7.1% to only –17% over the same range. Even on more challenging non-orthogonal tasks such as Puzzle and Rank, RL mitigates collapse far more effectively than SFT. These findings demonstrate that the benefits of RL-based training are **scalable and robust**, extending reliably to larger merging ensembles of 3–5 models.
>
> ||Average|||||
> |-|-|-|-|-|-|
> |Number of Merged Models|1|2|3|4|5|
> |SFT|61.5|49.9（-19%）|40.6（-34%）|39.9（-35%）|37.5（-69%）|
> |RL|72.2|67.1（-7.1%）|64.5（-11%）|62.6（-13%）|60.2（-17%）|
>
> ||Math|||||
> |-|-|-|-|-|-|
> |Number of Merged Models|1|2|3|4|5|
> |SFT|61.9|56.8(-8.3%)|56.3  (-9.1%)|55.2（-10.8%）|56.0（-9.5%）|
> |RL|64.6|63.3(-2.0%)|62.9（-2.7%）|62.3（-3.6%）|61.3（-5.2%）|
>
> ||Code|||||
> |-|-|-|-|-|-|
> |Number of Merged Models|1|2|3|4|5|
> |SFT|60.5|58.0（-4.1%）|55.6（-8.1%）|55.3（-8.8%）|56.2（-7.0%）|
> |RL|65.6|64.3（-2.0%）|63.4（-3.4%）|63.1（-3.8%）|62.4（-5.0%）|
>
> ||IF|||||
> |-|-|-|-|-|-|
> |Number of Merged Models|1|2|3|4|5|
> |SFT|63.9|47.5（-25%）|45.8（-28.3%）|45.5（-28.8）|45.1（-29%）|
> |RL|90.0|90.0（-0%）|89.5（-0.5%）|89.1（-1.0%）|87.8（-2.5%）|
>
> ||Puzzle|||||
> |-|-|-|-|-|-|
> |Model Merging Number|1|2|3|4|5|
> |SFT|86.2|35.8（-58%）|19.8（-77%）|7.5（-91%）|3.4（-96%）|
> |RL|85.2|64.6（-24%）|53.3（-27%）|43.9（-49%）|32.8（-62%）|
>
> ||Rank|||||
> |-|-|-|-|-|-|
> |Number of Merged Models|1|2|3|4|5|
> |SFT|52.8|51.3（-2.7%）|39.8（-26%）|33.6（-36%）|26.8（-51%）|
> |RL|55.7|53.1（-4.7%）|53.4（-3.9%）|54.8（-1.6%）|56.9（+2.1%）|
>
> ## statistical significance tests
> We thank the reviewer for raising this important point regarding statistical significance. To assess whether the observed performance differences are statistically significant, we conducted significance tests on the results reported in Table 1.
>
> As shown below, the performance improvements of RL-trained models over SFT-trained models are statistically significant (*p* < 0.05) for most tasks and merging methods, confirming that the observed improvements are unlikely to be due to random variation.
>
> ||Math|Code|IF|Puzzle|Rank|
> |-|-|-|-|-|-|
> |Averaging|0.016|0.058|0.013|0.009|0.213|
> |TIEs|0.009|0.016|0.002|0.006|0.100|
> |Arithmetic|0.033|0.011|0.007|0.005|0.045|
> |DARE|0.032|0.006|0.003|0.006|0.041|

---

> > ### Comment · Reviewer_W4Vs · 2025-11-28
> >
> > Thank you for the experimental results that have addressed my concerns. I will retain my positive evaluation.

---

### Official Review · Reviewer_uswu · 2025-10-28

**Soundness:** 4
**Presentation:** 3
**Contribution:** 4
**Rating:** 8
**Confidence:** 4

**Summary:**

This paper investigates the role of reinforcement learning (RL) versus supervised fine-tuning (SFT) in mitigating task conflicts during large language model (LLM) merging. The authors demonstrate that RL-trained models significantly outperform SFT-trained models in preserving task-specific performance after merging. Through theoretical and empirical analyses, the paper highlights three key mechanisms underlying RL's superiority: (1) the use of on-policy data, which reduces gradient magnitudes and parameter conflicts; (2) RL's adaptive optimization objective, which naturally decreases update magnitudes as training converges; and (3) joint optimization over positive and negative samples, which steers models toward unbiased task-specific parameter subspaces. The results offer valuable insights into building robust, scalable multi-task LLMs without retraining from scratch.

**Strengths:**

The paper addresses a novel and important problem—how different post-training paradigms influence task conflicts in LLM merging. The focus on RL as a more suitable paradigm for model merging is both creative and impactful.

The experiments are comprehensive, covering diverse tasks, multiple RL algorithms, and various merging strategies. Theoretical insights are rigorously derived and supported by empirical evidence.

The paper is well-organized, with clear explanations of concepts, methodologies, and results. Figures and tables effectively complement the narrative.

**Weaknesses:**

See questions.

**Questions:**

Could the authors provide more details about the experiments conducted on the Qwen model? Specifically, were there any unique observations regarding the performance of RL-based training on this model compared to others?

How do RL training hyperparameters, such as the coefficients for KL divergence and entropy, impact the performance of model merging?

In the GRPO algorithm, how does the choice of the parameter n (rollout number for each sample) influence the results? Were any ablation studies conducted to analyze its effect on task performance or parameter conflicts? What is the impact of batch size in the context of model merging? Does a larger batch size help stabilize training or reduce task conflicts more effectively?

Referring to the recent "LoRA Without Regret" paper by Thinking Machines, could the improved performance of RL in model merging also be attributed to the low-rank nature of its parameter updates? For example, does RL inherently constrain updates to a lower-rank subspace, similar to LoRA, thereby reducing task conflicts and preserving task-specific knowledge better? If so, could this be explicitly analyzed or compared in the context of your experiments?

---

> ### Author Response · Authors · 2025-11-21
>
> ## Question1: Could the authors provide more details about the experiments conducted on the Qwen model? Specifically, were there any unique observations regarding the performance of RL-based training on this model compared to others?
> We thank the reviewer for the question. We conducted additional experiments using **Qwen2.5-3B** on the Math, Code, and IF tasks, and evaluated both SFT-based merging and RL-based merging using the TIEs method. The results are summarized below:
>
>
>
> |Task|SFT|SFT-Merging|Δ (%)|RL|RL-Merging|Δ (%)|
> |-|-|-|-|-|-|-|
> |Math|72.2|68.2|–5.5%|72.9|71.9|–1.4%|
> |Code|64.2|61.9|–3.6%|68.2|66.9|–1.9%|
> |IF|58.0|25.1|–56.7%|67.0|65.9|–1.6%|
> |Average|64.8|51.7|–20.2%|69.4|68.2|–1.7%|
>
> As shown in the table, **RL-based training consistently preserves model performance under merging**, while SFT-based merging results in significantly larger degradation, especially on the IF task. This trend is consistent with our findings on other base models. A noteworthy observation is that **Qwen2.5-3B exhibits even smaller RL-merging degradation than Llama3.2-3B** under identical experimental settings. This suggests that the **effectiveness and stability of merging may also depend on the underlying capability and robustness of the base model.**
>
> ## Question4: Referring to the recent "LoRA Without Regret" paper by Thinking Machines, could the improved performance of RL in model merging also be attributed to the low-rank nature of its parameter updates? For example, does RL inherently constrain updates to a lower-rank subspace, similar to LoRA, thereby reducing task conflicts and preserving task-specific knowledge better? If so, could this be explicitly analyzed or compared in the context of your experiments?
>
>
>
> We appreciate the reviewer’s insightful question regarding the potential relationship between the rank of parameter updates and model merging performance. We conducted a preliminary analysis to investigate whether reinforcement learning (RL) inherently produces lower-rank parameter updates compared to supervised fine-tuning (SFT). However, our findings suggest that the improved merging performance of RL models is not primarily due to lower-rank updates.
>
> Specifically, we examined the rank characteristics of the updated parameters. To assess whether the updates are low-rank, we define the **effective rank** as the number of singular values (after singular value decomposition) that are larger than 10% of the largest singular value. The **effective rank ratio** is then computed as the effective rank divided by the total rank. For each model, we calculate the effective rank ratio for all parameter matrices with rank greater than 1 and report the average effective rank ratio per model. The results for SFT- and RL-trained models on the *Math*, *Code*, and *IF* tasks are summarized below:
>
> ||Math|Code|IF|
> |-|-|-|-|
> |SFT|48.7%|44.5%|29.6%|
> |RL|51.1%|48.8%|43.7%|
>
> As shown, RL-trained models do not exhibit lower effective ranks compared to SFT-trained models; in fact, the effective rank ratios are slightly higher. Referring to the *“LoRA Without Regret”* blog by Thinking Machines, one of its key findings is that “LoRA fully matches the learning performance of FullFT when running policy gradient algorithms for reinforcement learning.” While we agree that this observation points to a deeper underlying mechanism, our analysis indicates that the rank of parameter updates is unlikely to be the cause. Exploring this connection further is beyond the scope of our current study, but it would be an interesting direction for future research.
>
> ## Question2&3
> We are currently conducting relevant experiments. Due to time and resource constraints, the experiments have not yet been completed. We will provide updates as soon as the experiments are finished.

---

> > ### Author Response · Authors · 2025-12-03
> >
> > ## To Question "How do RL training hyperparameters, such as the coefficients for KL divergence and entropy, impact the performance of model merging?"
> > We thank the reviewer for the insightful question. To assess the impact of RL training hyperparameters—specifically the KL-divergence coefficient and the entropy coefficient—we conducted controlled experiments on **Qwen2.5-3B** across the Math, Code, and IF tasks. The results are summarized in the tables below. Overall, we observe that these hyperparameters **do influence the absolute performance of the RL-trained models**, particularly the entropy coefficient, which regulates exploration. However, their impact on **model merging performance (i.e., the degradation after TIEs merging)** is relatively minor. Across all tested settings, the post-merging performance drop remains consistently small (typically between –1% and –2.4%).
> >
> > |KL|Math|||Code|||IF|||
> > |-|-|-|-|-|-|-|-|-|-|
> > ||RL|TIEs|$\Delta$|RL|TIEs|$\Delta$|RL|TIEs|$\Delta$|
> > |0|66.9|66.2|-1.0\%|70.1|69.1|-1.4\%|67.2|66.2|-1.5\%|
> > |1e-3|72.9|71.9 |-1.4\%|68.2|66.9 |-1.9\%|67.0|65.9|-1.6\%|
> > |2e-3|72.5|71.5|-1.5\%|70.0|68.4|-2.3\%|67.5|66.1|-2.1\%|
> >
> >
> >
> >
> >
> > |Entropy|Math|||Code|||IF|||
> > |-|-|-|-|-|-|-|-|-|-|
> > ||RL|TIEs|$\Delta$|RL|TIEs|$\Delta$|RL|TIEs|$\Delta$|
> > |0|72.9|71.9 |-1.4\%|68.2|66.9 |-1.9\%|67.0|65.9|-1.6\%|
> > |1e-3|71.5|70.8|-1.0\%|71.3|69.6|-2.4\%|69.5|68.2|-1.8\%|
> >
> >
> >
> >
> >
> > ### To Question "In the GRPO algorithm, how does the choice of the parameter n (rollout number for each sample) influence the results? Were any ablation studies conducted to analyze its effect on task performance or parameter conflicts? What is the impact of batch size in the context of model merging? Does a larger batch size help stabilize training or reduce task conflicts more effectively?"
> > We appreciate the reviewer’s insightful question. To assess the impact of the GRPO rollout number nn and the RL batch size on task performance and model merging stability, we conducted ablation experiments on Qwen2.5 across the Math, Code, and IF tasks. The results are provided in the tables below. A larger batch size generally improves the **absolute RL performance**, but its effect on **post-merging degradation** is modest. The merging-induced drop remains within a narrow range (–1.4% to –2.2%), suggesting that batch size has **limited influence** on reducing cross-task parameter conflicts or improving merging robustness. Increasing the rollout number tends to **improve the RL model’s standalone performance**, as expected from higher-quality advantage estimates. However, similar to batch size, the impact on **merging stability** remains limited: the degradation after TIEs merging stays small across all rollout settings.
> >
> > |Batch size|Math|||Code|||IF|||
> > |-|-|-|-|-|-|-|-|-|-|
> > ||RL|TIEs|$\Delta$|RL|TIEs|$\Delta$|RL|TIEs|$\Delta$|
> > |128|73.2|71.7|-2.0\%|68.3|66.9|-2.0\%|65.8|64.4|-2.1\%|
> > |256|72.9|71.9 |-1.4\%|68.2|66.9 |-1.9\%|67.0|65.9|-1.6\%|
> > |512|74.7|73.1|-2.2\%|69.7|68.7|-1.4\%|72.8|71.8|-1.4\%|
> >
> >
> >
> > |Rollout number|Math|||Code|||IF|||
> > |-|-|-|-|-|-|-|-|-|-|
> > ||RL|TIEs|$\Delta$|RL|TIEs|$\Delta$|RL|TIEs|$\Delta$|
> > |4|71.8|70.4|-1.9\%|69.3|67.6|-2.4\%|65.4|64.3|-1.7\%|
> > |8|72.9|71.9 |-1.4\%|68.2|66.9 |-1.9\%|67.0|65.9|-1.6\%|
> > |16|74.6|73.0|-2.1\%|71.4|69.8|-2.5\%|69.2|68.5|-1.0\%|

---

### Official Review · Reviewer_ojWS · 2025-11-01

**Soundness:** 3
**Presentation:** 3
**Contribution:** 3
**Rating:** 6
**Confidence:** 3

**Summary:**

This paper addresses a crucial question in Large Language Models (LLMs): how the training paradigm, specifically Supervised Fine-Tuning (SFT) and Reinforcement Learning (RL), impacts the efficiency of model merging. Model merging aims to merge multiple specialized models into a single model, which is often plagued by multi-task conflicts that lead to performance degradation. The paper demonstrates that RL-trained models preserve performance better after merging compared to SFT-trained models.

**Strengths:**

+  Novel problem: The paper identifies a fundamental and significant question. The finding that RL can mitigate task conflicts is a major contribution, shifting the researcher's focus from post-hoc merging strategies to the pre-hoc training paradigm.

 +  Good empirical support: The problem is validated across 5 tasks , 4 merging methods (Table 1) , 3 RL algorithms (Figure 2) and 3 base models (Figure 3), almost fully proves the robustness of the core findings.

+  Deep insight: It not only proves RL is better, but also explore why. The three analysis (on-policy , RL objective , pos/neg samples) is insightful and well-supported by ablations. The performance landscape visualization (Figure 4) shows SFT models are "brittle" in task-specific directions while RL models are ''robust``.

**Weaknesses:**

-  In Table 1, the "Puzzle" task shows a very large performance drop even for RL-trained models (e.g., -56% for Averaging, -24% for TIEs). While this is still better than SFT's degradation (-65% to -56%), it is a much starker drop than for other tasks (like Code or IF). The paper notes this but doesn't fully explore why the Puzzle task remains a sharp drop for both paradigms. Does this suggest a limitation of RL's conflict mitigation, or is there a unique property of the logical puzzle task (e.g., requiring very sparse, specific parameter activations) that makes it an outlier?

- Attribution of Norm Reduction (SFT & RFT & RL):In Section 4.1 and Table 2, the update norms are SFT (6.50) > RFT (2.36) > RL (0.78) for the Math task. The author says that on-policy data helps regulate the norm (SFT vs. RFT). However, the gap between RFT (on-policy SFT) and RL (on-policy RL) is also very large (2.36 vs. 0.78). This suggests that the RL objective is also a major contributor to the small norm in Table 2. The paper presents these as separate sections (4.1 and 4.2), but the experiment in 4.1 seems to conflate both effects. The argument could be strengthened by clarifying how much of the norm reduction is due to on-policy data vs. the RL objective.

- The paper theorizes in Section 4.2 that the advantage A vanishes as the model converges. It would be good to see an empirical plot of the average reward r and/or average absolute advantage ∣A∣ over training steps. This would link the theory to the experimental results, strengthening this hypothesis.

**Questions:**

Q1. Why does the Puzzle task in Table 1 show severe degradation (-24% to -56%) even for RL models? Does this imply a unique property of this task's parameter subspace that makes it resistant to the "orthogonalization" effect of RL?

Q2. The norm for RL (0.78 for Math) is much smaller than for RFT (2.36). Since both are on-policy, is this difference entirely attributable to the RL objective from Section 4.2? Could the authors provide a clearer quantitative attribution of how much of the total norm reduction (from SFT's 6.50) comes from "on-policy data" (SFT to RFT) versus the "RL objective" (RFT to RL)?

Q3. Did the authors record the average reward and/or average absolute advantage ∣A∣ over training? A plot showing reward saturation and ∣A∣ decaying to 0 would be a powerful empirical validation for the "enough is as good as a feast" hypothesis.

Q4. Does the "RL-Pos" variant (learning only from positive samples) also result in a larger update norm (just like in Table 2) or a faster-growing conflict norm (like in Figure 6) compared to full RL?

**Details Of Ethics Concerns:**

No ethical issues identified.

---

> ### Author Response · Authors · 2025-11-21
>
> ## Question1: Why does the Puzzle task in Table 1 show severe degradation (-24% to -56%) even for RL models? Does this imply a unique property of this task's parameter subspace that makes it resistant to the "orthogonalization" effect of RL?
>
>
> We appreciate the reviewer’s insightful question. The Puzzle task indeed exhibits larger degradation compared with other tasks, even under RL-based merging. This behavior arises because the Puzzle task is **not orthogonal** to the other tasks being merged under the base model. Knight-and-Knave puzzles share substantial reasoning mechanisms—especially contradiction-driven logical elimination—with several other reasoning tasks. As a result, their parameter subspace is more entangled and thus more susceptible to interference during merging.
>
> To illustrate this, RL-trained Puzzle models typically rely on a **recursive contradiction-checking loop** (e.g., “assume X → propagate implications → detect contradiction → revise”). In contrast, RL-trained Math models tend to adopt a more **enumerative case-analysis strategy** (e.g., “case A / case B / …”). Although distinct, these reasoning styles partially overlap, which increases cross-task interference during the merging process. We provide examples below to demonstrate these different reasoning patterns.（Due to length limit, the example will be attached in a separate reply.
>
> Importantly, even in this challenging non-orthogonal setting, **RL still substantially reduces degradation relative to SFT**, indicating that RL-based merging remains more robust even for tasks with highly entangled reasoning structures.
>
> ## Question2: The norm for RL (0.78 for Math) is much smaller than for RFT (2.36). Since both are on-policy, is this difference entirely attributable to the RL objective from Section 4.2? Could the authors provide a clearer quantitative attribution of how much of the total norm reduction (from SFT's 6.50) comes from "on-policy data" (SFT to RFT) versus the "RL objective" (RFT to RL)?
>
>
> We thank the reviewer for this thoughtful question. To better disentangle the contributions of **on-policy data** and the **RL objective**, we conducted an additional controlled experiment. Specifically, we introduced an intermediate variant that we call **“On-policy RFT”**. This method repeatedly collects on-policy rollouts, filters responses with reward = 1, and trains solely on these filtered samples—**without** applying any RL optimization objective. Thus, it isolates the effect of on-policy data while holding the training objective fixed.
>
> The results are shown in the table below:
>
> ||Math|Code|IF|
> |-|-|-|-|
> |SFT|6.50 |7.75 |4.83|
> |RFT|2.36 |2.17|1.70|
> |on-policy RFT|1.45|1.31|1.13|
> |RL|0.78 |0.71|0.64|
>
> From these numbers, we observe:
>
> * The reduction from **SFT → RFT → On-policy RFT** accounts for **approximately 85%** of the total norm reduction.This reflects the dominant role of *on-policy data alignment*.
> * The additional reduction from **On-policy RFT → RL** accounts for the remaining **~15%**, which captures the contribution of the *RL optimization objective* itself.
>
> Therefore, while both on-policy data and the RL objective reduce parameter-space movement, the **majority of the effect comes from on-policy data**, with the **RL objective providing a further, smaller but consistent refinement** that leads to the lowest norm values
>
> ## Question3: Did the authors record the average reward and/or average absolute advantage ∣A∣ over training? A plot showing reward saturation and ∣A∣ decaying to 0 would be a powerful empirical validation for the "enough is as good as a feast" hypothesis.
> Thank you for the thoughtful question. Yes—we recorded both the average reward ® and the average absolute advantage (|A|) throughout training. Because figures cannot be included in the rebuttal, we report representative checkpoints below. In both tasks, as reward increases and saturates, |A| declines toward a small value, consistent with the “enough is as good as a feast” hypothesis.
>
> * Code
>
> |Training Steps|10|20|30|40|50|60|70|80|90|
> |-|-|-|-|-|-|-|-|-|-|
> |r|0.71|0.75|0.81|0.83|0.88|0.90|0.91|0.93|0.94|
> |\|A\||0.43|0.35|0.24|0.22|0.14|0.12|0.12|0.09|0.08|
>
> * IF
>
> |Training Steps |10|20|30|40|50|60|70|80|90|100|110|120|130|140|150|160|
> |-|-|-|-|-|-|-|-|-|-|-|-|-|-|-|-|-|
> |r|0.60|0.62|0.68|0.77|0.74|0.79|0.79|0.85|0.88|0.90|0.89|0.88|0.92|0.88|0.86|0.91|
> |\|A\||0.54|0.47|0.44|0.35|0.36|0.28|0.28|0.21|0.19|0.18|0.18|0.14|0.12|0.10|0.13|0.11|

---

> > ### Author Response · Authors · 2025-11-21
> >
> > ## Question4:  Does the "RL-Pos" variant (learning only from positive samples) also result in a larger update norm (just like in Table 2) or a faster-growing conflict norm (like in Figure 6) compared to full RL?
> > We thank the reviewer for the question. The **RL-Pos** variant does **not** exhibit a larger update norm or a faster-growing conflict norm compared with full RL. Because RL-Pos trains only on positive-reward samples, it still retains key components of RL—such as on-policy data collection and advantage estimation (with negative-sample advantages masked)—which helps maintain stability.
> >
> > The update norms for RL and RL-Pos are shown below:
> >
> > ||Math|Code|IF|
> > |-|-|-|-|
> > |RL|0.78|0.71|0.64|
> > |RL-Pos|0.59|0.52|0.66|
> >
> > We observe that RL-Pos produces update norms that are comparable to, and in two tasks even smaller than, those of full RL
> >
> > For conflict norms over training steps, the growth curves for RL and RL-Pos are:
> >
> > |Training Steps|0|16|32|48|64|80|96|112|128|
> > |-|-|-|-|-|-|-|-|-|-|
> > |RL |0|1.1|2.3|3.9|5.8|8.0|10.1|11.9|14.1|
> > |RL-Pos|0|0.9|2.2|4.0|6.2|8.3|10.5|12.0|13.0|
> >
> > Overall, RL-Pos follows a **very similar conflict-norm trajectory** to full RL and does **not** demonstrate the sharp increases seen in RFT in Figure 6.

---

> > > ### Author Response · Authors · 2025-11-21
> > >
> > > The example of Question 2 is as follows:
> > > ||Query |solution|
> > > |-|-|-|
> > > |Puzzle|You meet 2 inhabitants: Zoey, and Oliver.  Zoey remarked, \"Oliver is not a knight\". Oliver stated, \"Oliver is a knight if and only if Zoey is a knave\".  So who is a knight and who is a knave?|To solve this problem, we need to use the rules of the island: Knights always tell the truth, and knaves always lie.\n\n 1. Zoey says, \"Oliver is not a knight.\" Since Zoey is a knave, she is lying. Therefore, Zoey must be a knave.\n\n 2. Oliver says, \"Oliver is a knight if and only if Zoey is a knave.\" Since Zoey is a knave, Oliver is saying that he is a knight. Since Oliver is a knight, he is telling the truth.\n\n 3. Now, let's look at Zoey again. She said, \"Oliver is not a knight.\" Since Oliver is a knight, Zoey is lying. Therefore, Zoey is a knave.\n\n 4. Oliver said, \"Oliver is a knight if and only if Zoey is a knave.\" Since Oliver is a knight, Zoey must be a knave. We already know that Zoey is a knave, so Oliver is correct.\n\n 5. Since Oliver is a knight, and knights always tell the truth, Oliver is telling the truth when he says, \"Oliver is a knight if and only if Zoey is a knave.\" Therefore, Oliver is a knight.\n\n 6. Zoey said, \"Oliver is not a knight.\" Since Oliver is a knight, Zoey is lying. Therefore, Zoey is a knave.\n\n 7. Oliver said, \"Oliver is a knight if and only if Zoey is a knave.\" Since Oliver is a knight, Zoey must be a knave. We already know that Zoey is a knave, so Oliver is correct.\n\n 8. Now, let's look at Zoey again. She said, \"Oliver is not a knight.\" Since Oliver is a knight, Zoey is lying. Therefore, Zoey is a knave.\n\n 9. Oliver said, \"Oliver is a knight if and only if Zoey is a knave.\" Since Oliver is a knight, Zoey must be a knave. We already know that Zoey is a knave, so Oliver is correct.\n\n 10. Since Oliver is a knight, and knights always tell the truth, Oliver is telling the truth when he says, \"Oliver is a knight if and only if Zoey is a knave.\" Therefore, Oliver is a knight.\n\n 11. Zoey said, \"Oliver is not a knight.\" Since Oliver is a knight, Zoey is lying. Therefore, Zoey is a knave.\n\n 12. Oliver said, \"Oliver is a knight if and only if Zoey is a knave.\" Since Oliver is a knight, Zoey must be a knave. We already know that Zoey is a knave, so Oliver is correct.\n\n 13. Therefore, Oliver is a knight, and Zoey is a knave.\n\nCONCLUSION:\n(1) Zoey is a knave.\n(2) Oliver is a knight.|
> > > |Math|You meet 2 inhabitants: Zoey, and Oliver.  Zoey remarked, \"Oliver is not a knight\". Oliver stated, \"Oliver is a knight if and only if Zoey is a knave\".  So who is a knight and who is a knave?|To solve this problem, we need to analyze the given statements and use the properties of knights and knaves to deduce the identity of each character.\n\n 1. **Zoey's statement**: \"Oliver is not a knight\".\n   - If Zoey is a knight, then this statement is true (knights tell the truth).\n   - If Zoey is a knave, then this statement is false (knaves lie).\n\n 2. **Oliver's statement**: \"Oliver is a knight if and only if Zoey is a knave\".\n   - If Oliver is a knight, then this statement is true (knights tell the truth).\n   - If Oliver is a knave, then this statement is false (knaves lie).\n\n Let's analyze the possible scenarios:\n\n **Scenario 1: If Zoey is a knight:**\n- Zoey's statement is true (Oliver is not a knight).\n- Oliver's statement is true (Oliver is a knight if and only if Zoey is a knave, and since Zoey is a knight, Oliver is not a knight, which matches the statement).\n\n **Scenario 2: If Zoey is a knave:**\n- Zoey's statement is false (Oliver is a knight).\n- Oliver's statement is false (Oliver is a knight if and only if Zoey is a knave, but Zoey is a knave, so Oliver should be a knight, which contradicts the statement).\n\n From these scenarios, we can see that if Zoey is a knight, both statements are true, and if Zoey is a knave, both statements are false. This means that Zoey must be a knight.\n\n Now, let's analyze Oliver's statement further:\n\n - If Oliver is a knight, then his statement is true (knights tell the truth), and it implies that Zoey is a knave.\n - If Oliver is a knave, then his statement is false (knaves lie), and it implies that Zoey is a knave.\n\n Given that Zoey is a knight, Oliver's statement implies that Oliver is a knave (since Zoey is a knight, Oliver must be a knave for his statement to be true).\n\n Therefore, the identities are:\n\n- **Zoey**: Knight\n- **Oliver**: Knave\n\nCONCLUSION:\n(1) Zoey is a knight.\n(2) Oliver is a knave.|

---

> ### Author Response · Authors · 2025-11-28
>
> Dear Reviewer ojWS,
>
> Thank you for your thoughtful and constructive reviews. We hope our responses have addressed the concerns you raised, and we sincerely appreciate the opportunity to further refine our work. If you have any additional questions or suggestions, we would be grateful to receive them—your feedback is extremely valuable to us.
>
> Thank you again for your time, effort, and insightful comments.
>
> ICLR 2026 Submission24847 Author

---

### Official Review · Reviewer_6WLa · 2025-11-03

**Soundness:** 2
**Presentation:** 3
**Contribution:** 2
**Rating:** 4
**Confidence:** 3

**Summary:**

The paper discovers a phenomenon that the model trained by RL is more suitable for merging than model trained by SFT. From three aspects, the gradient magnitude, conflict parameter magnitude and effect of positive and negative samples, the paper designs experiments and explains the reasons why merging RL models has less performance degeneration.

**Strengths:**

1. The paper is well written and the motivation is clear.
2. The experiment results are consistent with the proposed ideas.

**Weaknesses:**

The designed experiments in the paper can basically verify the proposed factors which affects the model merging. However, there still are some shortcomings. First, the largest number of tasks is 5. Then, the paper does not study the performance of merging model trained on Math and Coder and model trained on IF and Puzzle. It is not clear that whether the number of training data will influence the results. Furthermore, if we keep the gradient of SFT and RL the same, will it affect the results? Finally, if we train SFT in continuous labels, will it affect the results? Also, please see questions below.

**Questions:**

1. The number of merging models is up to 5. What will happen if the models is more?
2. Whether the number of training data affects the results is not studied. What will happen if merging a model trained on two datasets with a model trained on other two datasets?
3. The comparison between SFT and RL models’ gradient magnitude is not enough. What will happen if we keep them the same gradient magnitude?
4. If we train SFT with normalized continuous label like the advantages of RL, will it affect the results?

---

> ### Author Response · Authors · 2025-11-21
>
> ## Question 1: The number of merging models is up to 5. What will happen if the models is more?
> We appreciate the reviewer’s question regarding the scalability of model merging beyond five models. We would first like to clarify that using five models already exceeds the common practice in existing literature, where most studies merge fewer than five models. For example, Yu et al. [1] merge three task-specialized models (Math, Code, and IF), while Wang et al. [2] similarly focus on three domains (Math, Code, and Science).
>
> In our work, we further extend the diversity of merged capabilities by incorporating three additional open-resource SFT and RL datasets covering ***table reasoning, temporal reasoning, and generative reward modeling*** [3–5]. To more directly address the reviewer’s concern, we conduct additional experiments scaling the number of merged models up to **eight**, and report results on Math, Code, and IF tasks. The results are summarized in the following table.
>
> As shown, **RL-merged models degrade much more gracefully than SFT-merged models** as the number of merged models increases. This demonstrates that RL-based merging maintains its advantage when scaling to larger numbers of models.
>
> |Number of Merged Models||1|6|7|8|
> |-|-|-|-|-|-|
> |Math|SFT|61.9|54.2(-12.4%)|54.4(-12.1%)|54.0(-12.8)|
> ||RL|64.6|61.0(-5.6%)|61.2(-5.3%)|60.8(-5.9%)|
> |Code|SFT|60.5|55.3(-8.8%)|55.0(-9.1%)|55.0(-9.1%)|
> ||RL|65.6|62.0(-5.5%）|62.0(-5.5%）|62.2(-5.2%）|
> |IF|SFT|63.9|44.6(-30.3%)|44.0(-31.1%)|44.2(-30.8%)|
> ||RL|90.0|86.8(-3.6%)|86.0(-4.4%)|85.6(-4.9%)|
>
>
>
> [1] Yu, Le, et al. "Language models are super mario: Absorbing abilities from homologous models as a free lunch." *Forty-first International Conference on Machine Learning*. 2024.
>
> [2] Wang, Yuanyi, et al. "Model Merging Scaling Laws in Large Language Models." *arXiv preprint arXiv:2509.24244* (2025).
>
> [3] Yang, Zheyuan, et al. "Table-r1: Inference-time scaling for table reasoning." *arXiv preprint arXiv:2505.23621* (2025).
>
> [4] Bazaga, Adrián, et al. "Learning to Reason Over Time: Timeline Self-Reflection for Improved Temporal Reasoning in Language Models." *arXiv preprint arXiv:2504.05258* (2025).
>
> [5] Chen, Nuo, et al. "Judgelrm: Large reasoning models as a judge." *arXiv preprint arXiv:2504.00050* (2025).
>
> ## Question 2: Whether the number of training data affects the results is not studied. What will happen if merging a model trained on two datasets with a model trained on other two datasets?
> We thank the reviewer for raising this important question regarding the impact of training data volume and diversity on model merging performance. We first clarify that it is standard practice in model merging research to train each base model on a single dataset or task. This design simplifies the analysis of capability transfer and isolates the contribution of each source model. Nevertheless, to directly address the reviewer’s concern, we conducted an additional experiment involving **multi-dataset training**.
>
> Specifically, we trained one model jointly on **Math + Code**, and another model jointly on **IF + Puzzle**. We then merged the two models using the TIEs merging method. The results across all four tasks are reported below.
>
> As shown in the table, **RL-based merging  preserves more performance compared with SFT-based merging**. This indicates that the advantages of RL in merging are robust to changes in dataset composition.
>
> ||Math|Code|IF|Puzzle|
> |-|-|-|-|-|
> |SFT|61.6|55.4|64.4|62.8|
> ||58.2(-5.5%)|51.9(-6.3%)|49.0(-23.9%)|23.8(-37.9%)|
> |RL|63.3|63.9|89.9|83.2|
> ||61.8(-2.4%)|62.1(-2.8%)|89.3(-0.7%)|70.4(-15.4%)|
>
> ## Question 3: The comparison between SFT and RL models’ gradient magnitude is not enough. What will happen if we keep them the same gradient magnitude?”
>
> Thanks for the reviewer’s thoughtful question. The observed differences in gradient magnitudes between SFT and RL models arise naturally from the distinct training data distributions and optimization algorithms used in each paradigm.
>
> During model training, the parameter update can be expressed as $\Delta \theta$= $\sum_{t=0}^N \eta_t \cdot g_t$, where $t$ denotes the training step,$N$ is the total number of training steps, $\eta_t$​ is the learning rate at step $t$, and $g_t$​ represents the gradient at step $t$. If the gradient magnitudes were artificially constrained to be identical across SFT and RL models, compensating adjustments (e.g., increasing the total number of training steps) would be required to achieve comparable parameter updates. Consequently, this modification would not fundamentally change the optimization trajectory or the resulting model performance.
>
> In summary, the gradient magnitude differences reflect intrinsic properties of the respective training processes rather than a controllable factor that independently determines performance outcomes.

---

> > ### Author Response · Authors · 2025-11-21
> >
> > ## Question 4:If we train SFT with normalized continuous label like the advantages of RL, will it affect the results?
> > We appreciate the reviewer’s thoughtful question. In principle, augmenting SFT with normalized continuous labels—similar to the advantage values used in RL—could influence training dynamics. Continuous labels would weight each training sample by its estimated contribution, which may help stabilize learning and reduce the magnitude of parameter updates.
> >
> > However, applying this idea in practice is still challenging. In RL the continuous advantage values are obtained through **multiple on-policy rollouts**, which provide an estimate of expected returns. Generating analogous continuous labels for SFT would require either human-annotated preference scores or multi-rollout evaluation for every sample, both of which are prohibitively expensive.

---

> ### Author Response · Authors · 2025-11-28
>
> Dear Reviewer 6WLa,
>
> Thank you for your thoughtful and constructive reviews. We hope our responses have addressed the concerns you raised, and we sincerely appreciate the opportunity to further refine our work. If you have any additional questions or suggestions, we would be grateful to receive them—your feedback is extremely valuable to us.
>
> Thank you again for your time, effort, and insightful comments.
>
> ICLR 2026 Submission24847 Author

---

### Author Response · Authors · 2025-12-03
**Summary of the Work and Rebuttal for Area Chair**

**Dear Area Chair**,

Thank you very much for taking the additional time to review our submission. We provide below a concise summary of our work and the key clarifications and results from the rebuttal period for your convenience.

---

### **Work Summary**
In this work, we aim to investigate the role of fine-tuning paradigms (SFT and RL) in model merging. Across extensive experiments, we identify a consistent and striking pattern:**Models fine-tuned with RL preserve their performance far better after merging than models fine-tuned with SFT.** To understand why RL provides this advantage, we perform a series of theoretical analyses and empirical studies. Our findings point to three core contributing mechanisms:

(1) On-policy training data in RL control the gradient updates in a smaller magnitude, reducing the risk of overwriting existing knowledge for other tasks in the model.

(2) The RL optimization objective, which favors “*enough is as good as a feast*”, progressively reduces the magnitude and the number of conflict parameter updates as the model converges.

(3) Joint optimization of positive and negative examples in RL** **steers the model towards an unbiased task-specific parameter subspace, ensuring robust performance while further preventing parameter conflicts.

---

### **Recognition from Reviewers**
We deeply appreciate the constructive and encouraging feedback from reviewers. Representative comments include:

**Research Problem**: "novel problem", "fundamental and significant question", "novel and important problem", "both creative and impactful"

**Analysis**: "deep insight", " insightful and well-supported by ablations", "theoretical insights are rigorously derived and supported by empirical evidence"

**Experiments:** "consistent with the proposed ideas", "good empirical support", "the experiments are comprehensive", "comprehensive empirical study"

---

### **Update in the Rebuttal Phase**
During the rebuttal phase, we directly addressed the reviewers’ questions and concerns. The key additions are summarized below:

**Expanded Experiments**

* We add broad new experiments demonstrating the RL's advantage in model merging, including adding 3 new taks for model merging (*reviewer* *6WLa*) ,  merging models tranined on multiple tasks (*reviewer* *6WLa*) ,  experiments on Qwen2.5-3B model (*reviewer uswu*), statistical significance tests (*reviewer* W4Vs), and detailed results of merging more models (*reviewer* W4Vs).
* We conducted extensive ablations showing that RL hyperparameters have limited impact on merging stability, including  coeffecients of KL divergenve and entropy (*reviewer uswu*),  rollout number and batch size (*reviewer uswu*).

**Additional Analyses**

* We added deeper analyses to support our theoretical claims, including the quantitive attribution of the total norm reduction, showing ~85% from on-policy data accounts and  ~15% from RL object(*reviewer* ojWS),  the change of averate reward and absolute advatange, illustrating the "*enough is as good as a feast*" property (*reviewer* ojWS), traning on positive samples only (RL-Pos) does not increase update norm and conflict norm (*reviewer* ojWS).
* We also introduced new analytical perspectives, including non-orthogonality of the Puzzle task explains its larger degradation during merging (*reviewer* ojWS), RL induces a higher-rank of parameter update (*reviewer* *usuw*).

**Additional Discussion**

* We clarified that enforcing identical gradient norms for SFT and RL would cause **s**ignificantly slower convergence for SFT (*reviewer* *6WLa*) and why adding normalized continuous labels to SFT could be beneficial in theory but is **impractical due to high annotation or rollout costs** (*reviewer* *6WLa*)

---

We hope our work contributes to a clearer understanding of the influence of different traning paradigms on model merging. We also aim for it to inspire future works on model merging and RL training.

Thank you for your time and careful consideration!

Best regards,

Authors of Submission 24847

---

### Meta-Review · Area_Chair_E9GA · 2026-01-15

**Summary:**

Acceptance is recommended. The paper establishes that Reinforcement Learning (RL) trained models are significantly more robust to merging-induced performance degradation than those trained with Supervised Fine-Tuning (SFT). Reviewers (Scores: 8, 6, 6, 4) found the extensive empirical validation—covering multiple tasks, merging methods, and base models—compelling. The theoretical analysis identifying three key factors (on-policy data, adaptive optimization, and joint positive/negative sample training) was praised as insightful and well-supported by ablations.

**Reviewer Concerns:**

The authors successfully addressed most concerns during the rebuttal:

Scalability: New experiments showed RL's advantage holds when merging up to 8 models and across multiple datasets.

Mechanism: Additional ablations quantified the contribution of on-policy data vs. optimization objectives, strengthening the mechanistic claims.

Statistical Significance: Significance tests were provided to confirm the observed performance gaps.

**Reviewer Scores:**

Scores improved or were maintained at a positive level (8, 6, 6, 4), reflecting consensus on the paper's novelty and rigorous execution. The single negative review (4) acknowledged the clear motivation but raised scalability questions that were largely mitigated by the rebuttal data. The work offers a valuable shift in perspective from post-hoc merging strategies to pre-hoc training paradigms.

---

### Decision · Program_Chairs · 2026-01-26

Accept (Poster)